# D-REX: DIFFERENTIABLE REAL-TO-SIM-TO-REAL ENGINE FOR LEARNING DEXTEROUS GRASPING

**Haozhe Lou**[1,2]\*, **Mingtong Zhang**[1,2]\*, **Haoran Geng**[3], **Hanyang Zhou**[2], **Sicheng He**[1,2], **Zhiyuan Gao**[1,2]
**Siheng Zhao**[1,2], **Jiageng Mao**[1,2], **Pieter Abbeel**[3], **Jitendra Malik**[3], **Daniel Seita**[2], **Yue Wang**[1]

[1,2]Physical Superintelligence (PSI) Lab, University of Southern California
[2]Viterbi School of Engineering, University of Southern California
[3]Department of EECS, University of California, Berkeley

## ABSTRACT

Simulation provides a cost-effective and flexible platform for data generation and policy learning to develop robotic systems. However, bridging the gap between simulation and real-world dynamics remains a significant challenge, especially in physical parameter identification. In this work, we introduce a real-to-sim-to-real engine that leverages the Gaussian Splat representations to build a differentiable engine, enabling object mass identification from real-world visual observations and robot control signals, while enabling grasping policy learning simultaneously. Through optimizing the mass of the manipulated object, our method automatically builds high-fidelity and physically plausible digital twins. Additionally, we propose a novel approach to train force-aware grasping policies from limited data by transferring feasible human demonstrations into simulated robot demonstrations. Through comprehensive experiments, we demonstrate that our engine achieves accurate and robust performance in mass identification across various object geometries and mass values. Those optimized mass values facilitate force-aware policy learning, achieving superior and high performance in object grasping, effectively reducing the sim-to-real gap. Our code and project page is available at drex.github.io.

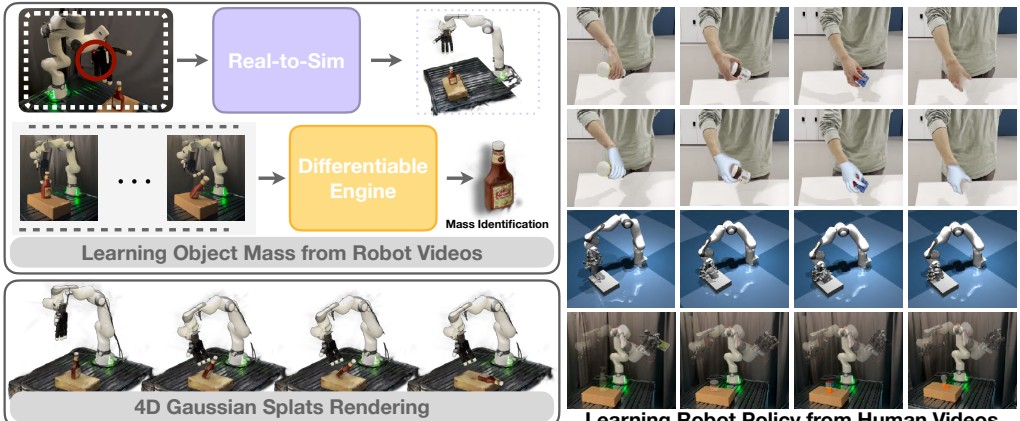

Figure 1: We present D-REX, a differentiable real-to-sim-to-real engine that enables 4D photorealistic rendering and physical simulation by identifying object mass from real-world visual observations and robot interaction data. D-REX reconstructs object geometry using Gaussian Splat representations and leverages a differentiable physics engine for end-to-end mass identification. The identified mass is then used to enable force-aware policy learning from human demonstrations, supporting robust grasping and sim-to-real transfer in dexterous grasping tasks.

## 1 INTRODUCTION

Simulation has become an essential platform for robotics, providing a cost-effective and scalable platform that reduces the reliance on extensive robotics expertise. Through reusable and controlled data generation, simulation has driven significant advancements in accelerating policy learning Akkaya

---

\*Equal contribution.

et al. (2019); Hafner et al. (2023); Chen et al. (2021); Agarwal et al. (2023); He et al. (2024); Ma et al. (2023). However, despite these benefits, replicating the visual realism and complex physical dynamics of the real world remains a significant challenge. High-fidelity physical simulations often demand specialized knowledge and complex modeling, which limits the scalability and robustness of simulation-based approaches for real-world deployment.

A long line of research has focused on bridging the sim-to-real gap, which arises when transferring models trained in simulation to real-world configurations. This gap remains a fundamental challenge in robotics. Simulation-based policies typically assume accurate knowledge and modeling of real-world configurations, including underlying physical parameters. However, differences between the estimated geometry and mass from visual observations and their real-world values increase the sim-to-real gap. Existing strategies to mitigate this gap include domain randomization Tobin et al. (2017); Sadeghi and Levine (2016); Peng et al. (2018a), which enhances robustness by varying simulation parameters, and system identification Hwangbo et al. (2019); Tan et al. (2018); Khalil et al. (2007), which refines simulation dynamics by calibrating with real-world observations. Advances in simulation fidelity Ho et al. (2020); Mittal et al. (2023) and domain adaptation Bousmalis et al. (2018); Ren et al. (2023); Chen et al. (2023) have further facilitated the transfer of models from simulation to reality in robotics applications. Complementary to these efforts, real-to-sim frameworks attempt to construct digital twins that replicate real-world geometry and dynamics with high precision Chen et al. (2024); Torne et al. (2024); Jiang et al. (2022). Nonetheless, building accurate digital twins typically requires integrating multiple approaches, such as geometric reconstruction and parameter identification. Despite these advances, achieving precise modeling from visual observations remains challenging for current real-to-sim methods.

This challenge is fundamentally tied to the problem of system identification—inferring physical parameters from visual observations to ensure simulated environments faithfully reflect real-world dynamics. Estimating object attributes and system dynamics from images is difficult, even with full system state access. While robust forward simulators Macklin et al. (2014) exist, their non-differentiability limits applicability to inverse problems. Surrogate gradient methods such as finite differences are commonly used Cranmer et al. (2020); Ramos et al. (2019); Wu et al. (2017), but scale poorly in high-dimensional settings. Recent progress in differentiable simulation improves learning efficiency. In particular, GradSim Jatavallabhula et al. (2021a) enables end-to-end differentiation from visual observations to object-level physical parameters. Inspired by this, our work optimizes object mass directly from video, enabling force-aware grasping policy learning conditioned on mass and substantially improving performance.

To address these challenges, we introduce D-REX in this paper, a differentiable real-to-sim-to-real framework that builds our simulation engine upon differentiable simulation Jatavallabhula et al. (2021a); Freeman et al. (2021); Müller et al. (2007); Macklin et al. (2016) and Gaussian Splat representations Kerbl et al. (2023). This differentiable engine enables object mass identification through visual observations and robot control signals in robot-object interactions. Additionally, we propose a novel learning-based method for dexterous manipulation, where we transfer human demonstrations into simulation-executable robot demonstrations, then utilize the proposed method to optimize the grasp position and force simultaneously.

Our main contributions include:
- A real-to-sim-to-real framework that enables end-to-end object mass identification through differentiable simulation from visual observations and robotic control signals.
- A novel approach to learn grasping policies from human demonstrations, conditioned on the identified object mass, that integrates position and force control to reduce the sim-to-real gap and achieve robust, high-performance grasping.
- Empirically, we show that identifying accurate mass with our differentiable framework and conditioning the policy on it improve dexterous grasping on challenging object.

## 2 RELATED WORKS

### 2.1 DIFFERENTIABLE PHYSICAL SIMULATION FOR ROBOTICS

The development of physical simulation enables efficient data generation and policy training for robotics Liu and Negrut (2021); Xu et al. (2021; 2022). Specifically, differentiable physical simulations have had great advancements recently, as they provide efficient gradients for policy learning. A popular approach is to develop a physical simulation with automatic differentiable program-

ming de Avila Belbute-Peres et al. (2018); Hu et al. (2019); Xu et al. (2022); Li et al. (2025a). Another line of work focuses on learning neural networks to approximate the real-world dynamics Li et al. (2019); Pfaff et al. (2020); Xian et al. (2021), which are inherently differentiable and suitable for applications in planning and control optimization. On the robotic application side, a variety of downstream tasks have been studied: fluid manipulation Xian et al. (2023), soft-body manipulation Huang et al. (2021), cloth manipulation Peng et al. (2024); Yu et al. (2023), and the co-optimization of soft robot morphology and control policies Bhatia et al. (2021) ; Li et al. (2023a); Li et al. (2025b). Notably, Jatavallabhula et al. (2021a) proposes to leverage differentiable multiphysics simulation for system identification from pixels. Our framework proposes a novel perspective to backpropagate the gradients obtained from visual observations for system identification through the differentiable approach, and enables dexterous manipulation policy learning from our real-to-sim results.

## 2.2 REAL-TO-SIM-TO-REAL TRANSFER

Real-to-sim enables the replication of real-world assets and dynamics in simulation, enhancing data-driven insights, optimization, and robotic capabilities. By capturing natural statistics, dynamic behaviors, and kinematic structures, it supports robust decision-making, efficient model training, and evaluation of complex scenarios. Several recent works exemplify these trends. Jiang et al. (2022) creates interactive digital twins of articulated objects for simulation. Chen et al. (2024); Mandi et al. (2024) generate articulated simulations from images, while Sundaresan et al. (2022) adapts parameters for deformable objects using point clouds. Neural Radiance Fields have also been applied to robotic tasks like manipulation and locomotion Kerr et al. (2022); Rashid et al. (2023); Zhou et al. (2023); Byravan et al. (2023); Wang et al. (2023), though often without accurate physical realism. More recent work Abou-Chakra et al. (2024); Zhang et al. (2024a); Kerr et al. (2024); Jiang et al. (2025); Zhobro et al. (2025); Abou-Chakra et al. (2025); Yang et al. (2025); Xie et al. (2023) leverages Gaussian Splats to construct digital twins from real-world visual input. Pfaff et al. (2025); Khalil et al. (2007) identify robot and payload parameters via joint-torque sensing. In contrast to prior work Zhu et al. (2025); Turpin et al. (2022), which often lacks integration with differentiable real-to-sim-to-real frameworks due to limitations in representation or simulation engines, our approach enables accurate object mass identification and policy learning in the simulation for direct real-world deployment.

## 2.3 LEARNING FROM HUMAN VIDEOS FOR ROBOTIC MANIPULATION

Human demonstration videos provide a scalable and semantically rich source for robotic manipulation. However, mapping human actions to robot control remains challenging due to differences in embodiment and sensing. Prior work addresses this gap by leveraging pre-trained visual representations Nair et al. (2022); Radosavovic et al. (2023); Ma et al. (2022), or by extracting intermediate cues such as affordances Bahl et al. (2023) and object-centric flow Xu et al. (2024), which are hard to perform fine-grained and dexterous manipulation. Others focus on 3D human motion estimation for skill transfer Shaw et al. (2023a); Patel et al. (2022); Peng et al. (2018b); Lum et al. (2025), which are often limited by the human and robot embodiment gap. Recent methods attempt to relax this constraint: Guzey et al. (2024) constructs reward functions from object tracking, and Singh et al. (2024) generates 3D hand-object trajectories from in-the-wild videos for retargeting. While promising, these approaches still struggle with generalizable policy learning from human videos. In contrast, our framework transfers human demonstrations into simulation-executable robotic demonstrations, enabling scalable policy learning with improved adaptability conditioned on object mass to improve the performance.

## 3 PROBLEM STATEMENT

We focus on the real-to-sim-to-real task, which aims to construct a simulation environment that closely mirrors real-world geometry, physics, and appearance. We assume access to several types of RGB videos: scene-centric video sequences, denoted as $\mathcal{I}_s$, which capture the static environment; object-centric videos, denoted as $\mathcal{I}_o$, which provide multiple views of the manipulated object to support accurate visual and geometric reconstruction; and human demonstration videos $\{\mathcal{I}_t\}_{t=1}^T$, which illustrate task execution. We also extract object trajectories from real-world robot rollouts, denoted as $\{s_t^{real}\}_{t=1}^T$, and simulate corresponding trajectories $\{s_t^{sim}\}_{t=1}^T$ within our framework.

The scene and object videos ($\mathcal{I}_s$, $\mathcal{I}_o$) are used to initialize the real-to-sim process via visual and geometric reconstruction. Trajectories from both real and simulated rollouts enable mass identification through a differentiable engine, producing an optimized object mass $m$ that is incorporated as a physical parameter in the simulator. Meanwhile, human demonstration videos $\{\mathcal{I}_t\}_{t=1}^T$ are translated

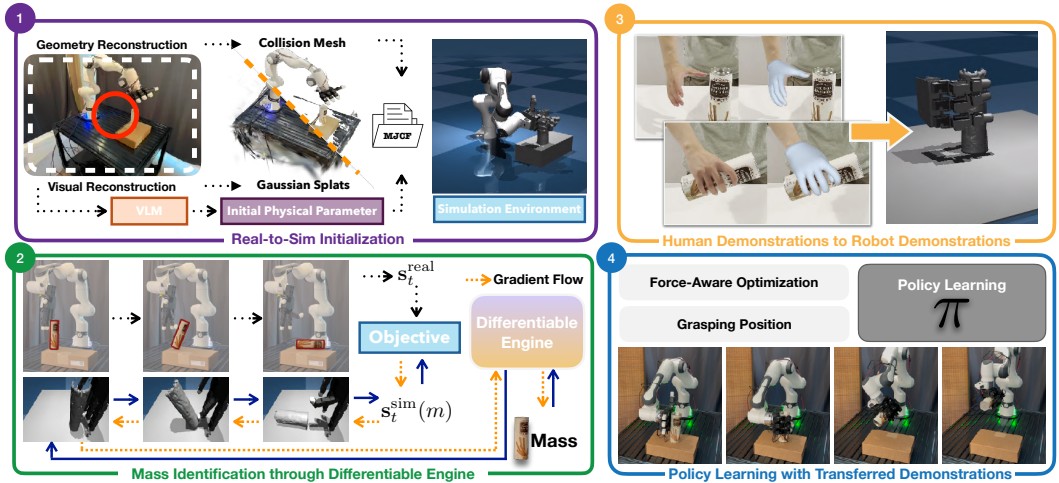

Figure 2: **Overview of our method.** Our approach consists of four components: (1) Real-to-Sim, (2) Mass Identification, (3) Learning from Human Demonstrations, and (4) Policy Learning. We begin by capturing videos of the scene and human demonstrations. Robotic actions are then executed in both simulation and the real world to identify object mass via our differentiable physics engine. Lastly, a manipulation policy is trained using the demonstrations and identified mass.

into robot-executable trajectories $\{\mathbf{A}_t\}_{t=1}^T$ using our proposed method and are used to train a force-aware manipulation policy $\pi$.

## 4 METHODS

We propose a real-to-sim-to-real framework that constructs accurate simulation environments and identifies object mass via system identification using a differentiable engine from visual observations and robot control signals, enabling robust policy learning and sim-to-real transfer. The framework is built on MuJoCo Todorov et al. (2012), a general-purpose physics simulator, the differentiable engine Brax Freeman et al. (2021), and Gradsim Jatavallabhula et al. (2021a). It operates in four steps shown in Figure 2: Real-to-Sim (Section 4.1), Mass Identification (Section 4.2), Learning from Human Demonstrations (Section 4.3), and Policy Learning (Section 4.4). First, the scene and object are reconstructed from RGB videos $\mathcal{I}_s$ and $\mathcal{I}_o$, capturing static environments and target objects. The output simulation is formalized as $\mathcal{S} = \{\mathcal{K}, \boldsymbol{\theta}\}$ in MJCF format, where $\mathcal{K}$ denotes collision meshes and $\boldsymbol{\theta}$ the physical parameters. Next, the framework executes consistent robotic actions in simulation and the real world, collecting $\{\mathbf{s}_t^{\text{real}}\}_{t=1}^T$ and $\{\mathbf{s}_t^{\text{sim}}\}_{t=1}^T$ to identify object mass $m$. Third, human demonstrations $\{\mathcal{I}_t\}_{t=1}^T$ are translated into robot-executable trajectories $\{\mathbf{A}_t\}_{t=1}^T$. Finally, these trajectories are used to train a policy in simulation, which is then deployed in the real world directly.

### 4.1 VISUAL AND GEOMETRIC RECONSTRUCTION

Our framework starts with reconstructing high-fidelity visual and geometric models of key elements in the manipulation environment with $\mathcal{I}_s$, including objects, robotic arms, dexterous hands and workspaces. This reconstruction ensures accurate representations of both collision geometry and visual appearance. To integrate these models into a differentiable simulation, we adopt the Gaussian Splat representation Kerbl et al. (2023); Huang et al. (2021), which enables photorealistic rendering and high-quality mesh generation for collision detection following Lou et al. (2024). Specifically, we process videos collected from mobile devices to train two ensembles of Gaussian primitives: one for collision geometry and another for visual appearance. Specifically, 2D Gaussian Splats with surface normal estimation Ye et al. (2024a) provide accurate geometry for simulation, while 3D Gaussian Splats ensure high-fidelity rendering. This process yields two complementary outputs: a collision mesh $\mathcal{K}$ and Gaussian particles $\mathcal{P}$. Additional details are provided in the Appendix.

### 4.2 PHYSICAL PARAMETER IDENTIFICATION FROM ROBOT-OBJECT INTERACTIONS

Accurate identification of physical parameters $\boldsymbol{\theta}$ is essential for constructing physically plausible simulations. We begin by using a Vision-Language Model Hurst et al. (2024) to generate an initial MJCF representation $\mathcal{S}$ from environment images and prompts Zhang et al. (2024b). While this approach provides a reasonable structural prior, parameters inferred solely from visual inputs often deviate from real-world values due to the lack of observable physical cues Asenov et al. (2020).

To address this, we focus on identifying the object mass, which is a key parameter in dynamics that can be reliably measured. Accurate mass identification improves simulation fidelity and enables robust policy learning. We choose a planar pushing task with a virtual fulcrum assumption to reduce frictional effects, and optimize mass $m$ to minimize the discrepancy between simulated and real-world trajectories Jatavallabhula et al. (2021a):

$$\min_{m>0} \ \mathcal{L}_{\text{traj}}(m) := \sum_{t=1}^{T} \big\| \mathbf{s}_t^{\text{sim}}(m) - \mathbf{s}_t^{\text{real}} \big\|_2^2, \tag{1}$$

where $\mathbf{s} = [\mathbf{p}, \mathbf{q}]^\top \in \mathbb{R}^7$ denotes the object's 6-DoF pose, consisting of position $\mathbf{p} \in \mathbb{R}^3$ and orientation represented as a unit quaternion $\mathbf{q} \in \mathbb{R}^4$. $\mathbf{s}_t^{\text{real}}$ is obtained by FoundationPose Wen et al. (2024) in the real world while $\mathbf{s}_t^{\text{sim}}(\mathbf{m})$ is obtained by executing the same actions in the simulation.

**Dynamics Modeling.** To simulate object motion, we adopt a standard rigid-body formulation of the Newton-Euler mechanism. Let $\mathbf{u}_t = [\mathbf{v}_t, \boldsymbol{\omega}_t]^\top$ denote the object's velocity at timestep $t$, where $\mathbf{v}_t$ and $\boldsymbol{\omega}_t$ are the linear and angular velocity components, respectively. We express the governing equation as the second order differential equation(ODE) Chen et al. (2019):

$$\mathbf{M}(\mathbf{s}_t, \mathbf{u}_t, m, \boldsymbol{\theta}) \, \dot{\mathbf{u}}_t \ = \ \mathbf{f}(\mathbf{s}_t, \mathbf{u}_t, \boldsymbol{\theta}), \tag{2}$$

where $\mathbf{M}$ is the mass-inertia matrix Baraff (1992) and $\mathbf{f}$ collects external and contact forces equation 3, gravity and torques. We adopt a compliant penalty-based contact model, parameterized by stiffness and damping $(k_e, k_d) \in \boldsymbol{\theta}$, which applies normal forces proportional to penetration depth and contact velocity Todorov et al. (2012); Erez et al. (2015):

$$\mathbf{f}_n(\mathbf{s}, \mathbf{u}_t, \boldsymbol{\theta}) \ = \ -\mathbf{n} \big( k_e \, C(\mathbf{s}) + k_d \, \dot{C}(\mathbf{u}) \big), \tag{3}$$

where $\mathbf{f}_n$ is the contact force, $\mathbf{n}$ is the contact normal, $C(\mathbf{s})$ the penetration depth, and $\dot{C}(\mathbf{u})$ is the derivative of $C(\mathbf{s})$. This contact model is differentiable and readily integrated into our simulation framework. In practice, we implement the dynamics using a discrete-time update Erez et al. (2015):

$$\mathbf{s}_{t+1}^{\text{sim}} = G\big(\mathbf{s}_t^{\text{sim}}, \mathbf{u}_t, m, \mathbf{f}_t\big), \quad t = 0, \ldots, T-1. \tag{4}$$

$\mathbf{f}_t$ are external forces at timestep $t$, including actuator impulses, gravity, and object-ground contacts.

**Differentiable Physics.** To optimize equation 1, we compute gradients of the simulated trajectory with respect to the object mass $m$. Following the discrete adjoint method from Jatavallabhula et al. (2021a), we adopt a semi-implicit Euler integration scheme for stability under contact dynamics. We couple kinematics from MjX/Brax Freeman et al. (2021) with rigid-body dynamics equation 2 and the contact model equation 3, forming a differentiable computation graph Hu et al. (2020).

**Semi-Implicit Euler Modeling.** The update function $G(\cdot)$ in equation 4 is implemented using a semi-implicit Euler integration scheme:

$$G\Big([\mathbf{s}_t, \, \mathbf{u}_t], m, \, \boldsymbol{\theta}\Big) = \begin{bmatrix} \mathbf{s}_t \ + \ \Delta t \, \mathbf{u}_{t+1} \\ \mathbf{u}_{t+1} \end{bmatrix} = \begin{bmatrix} \mathbf{s}_t \ + \ \Delta t \Big( \mathbf{u}_t + \Delta t \, \mathbf{M}^{-1}(\mathbf{s}_t, \mathbf{u}_t, m, \boldsymbol{\theta}) \, \mathbf{f}\big(\mathbf{s}_t, \, \mathbf{u}_t\big) \Big) \\ \mathbf{u}_t + \Delta t \, \mathbf{M}^{-1}(\mathbf{s}_t, \mathbf{u}_t, m, \boldsymbol{\theta}) \, \mathbf{f}\big(\mathbf{s}_t, \, \mathbf{u}_t\big) \end{bmatrix} \tag{5}$$

where $\Delta t$ is the integration timestep, and $f(\cdot)$ encapsulates both external and contact forces.

**Differentiable Real-to-Sim-to-Real Optimization.** We simulate the system starting from the initial condition $\mathbf{s}_0^{\text{sim}}$, and iteratively update the state via equation 4. To quantify the discrepancy between simulated and real-world trajectories, we define the trajectory loss between the simulated state $\mathbf{s}_t^{\text{sim}}$ and the corresponding real-world state $\mathbf{s}_t^{\text{real}}$ as:

$$\mathcal{L}_{\text{traj}}(m) = \sum_{t=0}^{T} \big\| \mathbf{s}_t^{\text{sim}} - \mathbf{s}_t^{\text{real}} \big\|_2^2. \tag{6}$$

This objective encourages the simulated trajectory, parameterized by mass $m$, to closely match the observed real-world dynamics over time. The gradient $\nabla_m \mathcal{L}_{\text{traj}}(m)$ is computed via automatic differentiation, using backpropagation as implemented in PyTorch Paszke et al. (2019), as follows:

$$\frac{\partial \mathcal{L}_{\text{traj}}}{\partial m} = \sum_{t=1}^{T} \frac{\partial \mathcal{L}_{\text{traj}}}{\partial \mathbf{s}_t^{\text{sim}}} \cdot \frac{\partial \mathbf{s}_t^{\text{sim}}}{\partial \mathbf{M}_t} \cdot \frac{\partial \mathbf{M}_t}{\partial m}. \tag{7}$$

Unlike system identification methods such as GradSim Jatavallabhula et al. (2021a), which rely on manually specified external forces, our approach supports end-to-end optimization by directly leveraging consistent robotic control signals in both simulation and the real world to model the external forces applied to the object. This creates a tight coupling between real-world and simulated trajectories, enabling us to capture contact dynamics through robot-object interactions.

Importantly, our method does not require ground-truth object mass or contact points. Object geometry and poses are obtained via Section 4.1, while actuator signals and robot-object interactions are derived from the MJCF kinematic model. These serve as inputs to our differentiable framework for accurate mass optimization. Additional modeling details and experiments are provided in the Appendix.

### 4.3 Transferring Human Demonstrations to Robot Demonstrations

After accurately modeling the scene and object, the next step is to collect real-world human demonstrations and transfer them into robot demonstrations for policy learning. Although learning directly from human demonstrations is intuitive, substantial differences between human and robotic hands complicate grasp interaction transfer, particularly due to varied object geometries and masses.

Our approach aims to transform human demonstrations captured from RGB video sequences $\{\mathcal{I}_t\}_{t=1}^T$ into executable robotic demonstrations within the simulation. Each video frame $\mathcal{I}_t$ is processed using HaMeR Pavlakos et al. (2024) and MCC-HO Wu et al. (2024a) to reconstruct detailed articulated models of the human hand and the manipulated object. At each timestep $t$, these methods output:

$$\mathbf{h}_t \in \mathrm{SE}(3) \times \mathbb{R}^{J_h}, \quad \mathbf{o}_t \in \mathrm{SE}(3), \tag{8}$$

where $\mathbf{h}_t$ encodes the 6-DoF wrist pose and finger joint angles ($J_h$), and $\mathbf{o}_t$ describes the object's 6-DoF pose. Subsequently, we employ Dex-Retargeting Qin et al. (2023) to map these human hand-object poses $\mathbf{h}_t, \mathbf{o}_t$ to the robotic hand with $J_r$ degrees of freedom. This produces robot actions:

$$\mathbf{A}_t = \mathcal{R}(\mathbf{h}_t, \mathbf{o}_t) \in \mathbb{R}^{J_r}, \tag{9}$$

where $\mathbf{A}_t$ represents target joint angles for robotic actuators. Given our assumption that the object geometry remains consistent between human demonstration and robotic manipulation, the resulting action set $\mathbf{A}_t$ directly serves as a data source for our policy learning.

### 4.4 Policy Learning with Transferred Robot Demonstrations

We initialize policy learning using robot demonstrations $\{\mathbf{A}_t\}_{t=1}^T$ described in Section 4.3. Each demonstration maps the reconstructed object's collision mesh vertices $\mathcal{K}$ as inputs to the corresponding robotic grasp pose. These observation-action pairs directly supervise training of the manipulation policy $\pi_\phi$, which maps object-centric observations to dexterous grasp configurations.

**Grasping Position Learning.** To capture an object's geometry and pose, we encode the vertices of its collision mesh $\mathcal{K}$ using positional encoding Tancik et al. (2020), forming the input to our policy. This policy conditions the observation $\mathbf{o}$ on the reconstructed collision mesh $\mathcal{K}$ and the identified mass $m$. Concretely, $\pi_\phi$ is a multi-head neural network that predicts dexterous hand joint positions $\hat{\mathbf{A}}$, contact-related rewards $\hat{\mathbf{r}}$, and a mass-related control force $\hat{\mathbf{f}}$.

$$\pi_\phi(\mathbf{o}) = \begin{bmatrix} \hat{\mathbf{A}} \\ \hat{\mathbf{r}} \\ \hat{\mathbf{f}} \end{bmatrix} \in \mathbb{R}^{19}, \quad \hat{\mathbf{f}} = \frac{m \cdot g}{n_{\mathrm{active}}}. \tag{10}$$

where $\hat{\mathbf{A}} \in \mathbb{R}^{16}$ denotes the predicted joint positions, $\hat{\mathbf{r}} \in \mathbb{R}^2$ represents the contact constraint, and $\hat{\mathbf{f}} \in \mathbb{R}$ denotes the grasping force constraint. The variable $n_{\mathrm{active}}$ indicates the number of active contacts between the robotic hand and the object. The network uses fully connected layers with ReLU activations, followed by a pooling layer. More details are in the Appendix.

**Force-Aware Optimization Design.** At training onset, we define a two-dimensional contact constraint $\hat{\mathbf{r}}$: one term encourages sustained contact during the rollout, the other ensures object retention at the end. The policy dynamically influences this constraint based on hand–object interactions through our simulation engine and the simulation asset $S$ from Section 4.1 and 4.2. It remains high when active contact points exceed a threshold $N_{\mathrm{min}}$ over the time horizon $H$:

$$\forall t \in [t_0, t_0 + H] : n_{\mathrm{active}}(t) \geq N_{\mathrm{min}}, \quad \mathbb{I}_{\mathrm{in\_hand}}(t) = \begin{cases} 1, & \text{if } n_{\mathrm{active}}(t) \geq 1, \\ 0, & \text{otherwise,} \end{cases} \tag{11}$$

We subsequently retrain the manipulation policy with the force-based constraint in 10, enabling adaptive force control that responds to object mass variations Vassiliadis et al. (2021); Zhang et al. (2025). This enhances the robustness of grasp poses learned from demonstrations, ensuring stability under diverse dynamics.

Traditional position-based policies replicate grasp poses from human demonstrations Patel et al. (2022); Lum et al. (2025); Chen et al. (2025a); Wan et al. (2023); Wei et al. (2025); Shaw et al. (2022) but overlook unobserved forces, particularly those countering gravity. Applying uniform forces across varying object masses often leads to instability. To address this, we propose a hybrid control framework that combines position and force control, using a prediction module conditioned on the optimized mass $m$. This jointly optimizes the policy parameters $\phi$ and grasping force, enabling more robust and physically grounded manipulation.

## 5 EXPERIMENTS

The objective of our experiments is to evaluate the performance of our system across the following key aspects: (1) Evaluate the effectiveness and robustness of mass identification using the differentiable engine across varying object geometries, densities, and categories. (2) Analyze how incorporating object mass affects policy learning performance, assessing the feasibility of force-based control. (3) Assess the effectiveness of learning grasping policies from transferred robot demonstrations and their direct sim-to-real deployment.

### 5.1 MASS IDENTIFICATION VIA OBJECT PUSHING

We evaluate our mass identification method through object pushing experiments by applying identical actions in both the real world and simulation. The resulting trajectories are used to optimize object mass via our differentiable engine, as detailed in Section 4.2. We assess the performance in two settings: (1) across objects with varying geometries, and (2) across replicas with identical geometry but different internal densities. Our method accurately recovers mass in both cases, demonstrating generalization to diverse shapes and sensitivity to subtle physical differences. The objects we used for mass identification are shown in Figure 3.

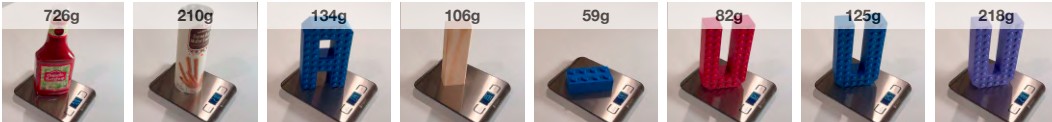

Figure 3: **Objects for Mass Identification.** We conduct experiments on mass identification across diverse object geometries and identical geometries with varying densities. Our method accurately estimates mass in both settings, demonstrating robustness to shape and density variations.

To evaluate robustness across shapes, sizes, and mass scales, we select a diverse set of objects for mass identification. This tests the pipeline's ability to generalize across varying contact geometries. As shown in Table 1, percentile errors range from 4.8% to 12.0%, demonstrating accurate mass optimization without object-specific tuning.

To isolate the effect of mass, we fabricate three replicas with identical geometry but varying internal densities $\rho$ using different 3D printing infill ratios. By keeping shape constant, any identification error reflects mass sensitivity. As shown in Table 2, mass is accurately identified with deviations under 13 grams, confirming the effectiveness of estimating physical parameters independent of geometry.

As shown in Figure 4, simulations using the optimized mass closely match real-world object dynamics, while those using an incorrect lighter mass deviate significantly. This demonstrates that accurate mass identification improves both the physical realism and visual quality of simulated rollouts.

| Object | Letter U | Letter A | Lego | Domino | Cookie | Ketchup |
|---|---|---|---|---|---|---|
| Inferred Mass (g) | 500 | 500 | 300 | 500 | 500 | 1000 |
| Identified Mass (g) | 110 | 145 | 53 | 117 | 200 | 667 |
| Ground Truth Mass (g) | 125 | 134 | 59 | 106 | 210 | 726 |
| Percentile Error (%) | 12.0 | 9.0 | 8.6 | 9.3 | 4.8 | 8.1 |

Table 1: Mass identification across diverse objects with varying shapes, sizes, and mass scales. The inferred mass is obtained from VLM as described in Section 4.2.

| Density | $\rho_1$ | $\rho_2$ | $\rho_3$ |
|---|---|---|---|
| Identified Mass (g) | 95 | 129 | 207 |
| Ground Truth Mass (g) | 82 | 125 | 218 |

Table 2: Mass identification across objects with identical geometry but varying densities.

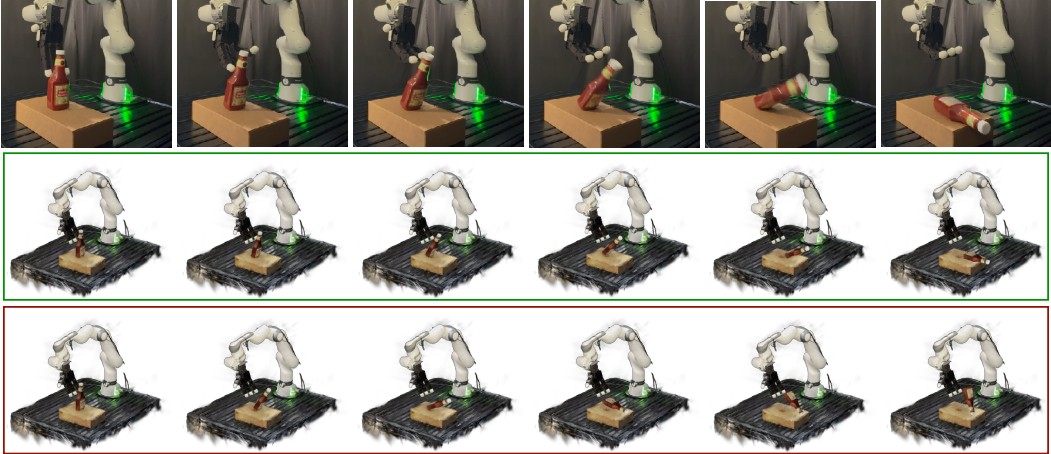

Figure 4: **Quantitative Results of Mass Identification.** We show the real-world object pushing (top) and render object trajectories using Gaussian Splats: simulated with optimized mass (middle), and simulated with a lighter mass (bottom), all using the same robot actions. The optimized mass closely reproduces real-world dynamics, reducing the sim-and-real gap with high visual fidelity.

| Train \ Eval | $\rho_1$ | $\rho_2$ | $\rho_3$ |
|---|---|---|---|
| $\rho_1$ | **75%** | 30% | 15% |
| $\rho_2$ | 40% | **80%** | 30% |
| $\rho_3$ | 15% | 40% | **95%** |

Table 3: Cross-evaluation of grasping policies trained on different object densities and evaluated across varying masses. Each cell shows the grasp success rates. Policies perform well only when the training and evaluation masses match.

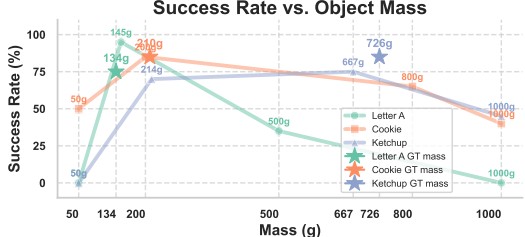

Figure 5: Grasping success rates across three objects with different mass values.

These experiments demonstrate that our differentiable Real-to-Sim-to-Real framework achieves accurate mass identification across both inter-object diversity and intra-object density variations.

## 5.2 Effectiveness of Force-Based Control through Grasping

In our grasping experiments, we evaluate how incorporating force-based constraints conditioned on object mass influences sim-to-real performance. This setup highlights the need for mass-aware force control and demonstrates the impact of accurate mass identification on policy success.

We first evaluate our grasping policy on three objects that share identical geometry and demonstrations but differ in mass. Each policy is trained with a specific object mass to assess the impact of mass-aware force control. As shown in Figure 6, policies perform well only when training and evaluation masses matched: the medium-mass policy succeeds on the medium object but fails on the heavier and lighter ones due to under- and over-applied force, respectively. Mass mismatches likewise lead to unstable grasps for the other two policies. Table 3 confirms this trend, with the highest success rate (80%) on the training mass, while performance drops to 40% and 30% on mismatched cases. These results highlight the importance of accurate mass conditioning for robust, reliable grasping.

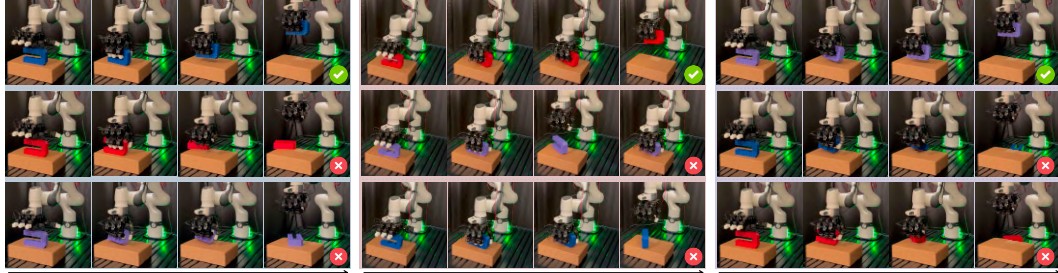

Figure 6: **Qualitative Results.** Left to right: policies trained on medium, light, and heavy objects. Only the mass-matched policy achieves stable grasps, while mismatched ones fail due to excessive or insufficient force, causing bounce-off for lighter objects or slippage for heavier ones.

Second, we evaluate whether policies conditioned on automatically identified mass can match the performance of those trained with object mass. As shown in Figure 5, success rates consistently peak at either the ground-truth or identified mass. Notably, policies using identified mass often match or even exceed those using ground-truth values, while substantially outperforming policies conditioned on arbitrary masses. These results underscore the effectiveness of our mass identification approach in enabling robust, force-aware grasping without requiring access to true mass values.

These experiments demonstrate that accurate mass is essential for effective force-aware grasping. Policies trained with object mass consistently outperform those trained on mismatched masses, and policies conditioned on automatically identified mass achieve comparable performance to those using ground-truth values. Together, these results validate our mass identification framework as a practical and reliable solution for enabling robust grasping without prior knowledge of object mass.

## 5.3 TABLETOP OBJECT GRASPING EXPERIMENTS

We compare our grasping policy against two baselines across various objects: (1) DexGraspNet 2.0 Zhang et al. (2024c), trained on large-scale simulation datasets, and (2) Human2Sim2Robot Lum et al. (2025), a recent real-to-sim-to-real method that learns dexterous manipulation policies from RGBD videos of human demonstrations. All use the collision mesh $\mathcal{K}$ generated by our real-to-sim framework as input. As shown in Figure 7, our method consistently outperforms the baselines across eight objects with diverse geometries and masses, achieving high success rates with substantially lower variance. While baseline performance degrades as object mass increases, our force-aware policy maintains stable, reliable grasps across the full range of object characteristics.

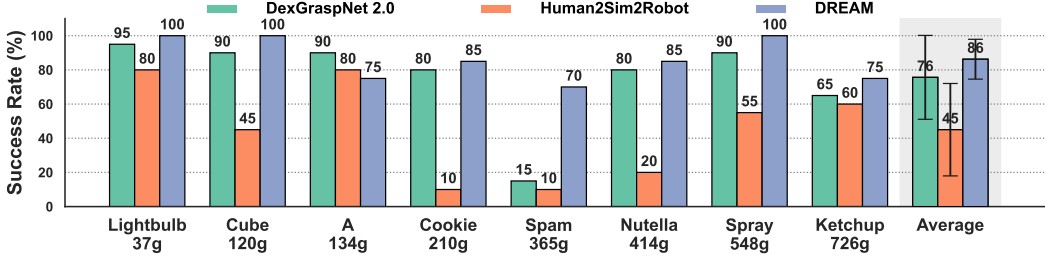

Figure 7: **Quantitative Results of Grasping Policies.** Grasp success rates across eight objects with varying geometries and mass values, with the average and standard deviation of each method.

Figure 8 presents qualitative results of our force-aware grasping policy across a range of objects. The top row captures the motion leading to the pre-grasp pose, while the bottom row displays the resulting post-grasp configurations. These examples demonstrate the policy's ability to consistently achieve stable and secure grasps under varying object geometries and mass values.

| Object | # Images | COLMAP+3DGS+2DGS runtime | Mesh vertices |
|--------|----------|--------------------------|---------------|
| Lego | 306 | 30 min | 47,266 |
| Ketchup | 337 | 35 min | 13,487 |
| Domino | 320 | 30 min | 35,445 |
| Cookie | 309 | 30 min | 25,682 |
| A | 306 | 30 min | 67,895 |
| U | 327 | 33 min | 15,322 |

Table 4: Offline reconstruction statistics for representative objects, including the number of input images, total runtime, and the number of vertices in the extracted mesh.

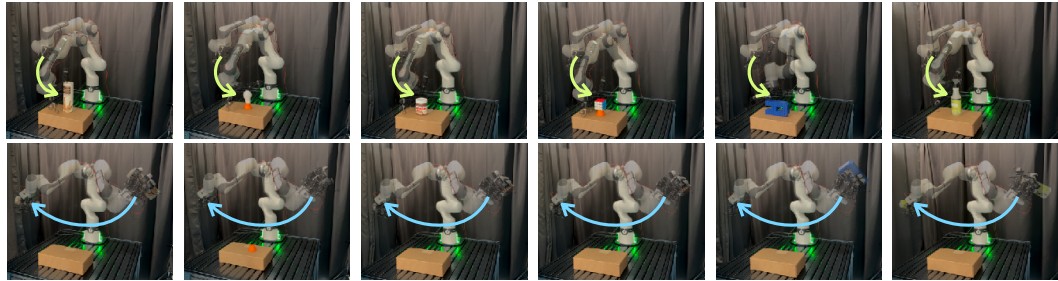

Figure 8: **Qualitative Results of Our Policy.** We evaluate our force-aware grasping policy across various objects. The first row illustrates the approach to the pre-grasp pose, while the second row shows two post-grasp positions, demonstrating that the policy achieves stable, secure grasps.

### 5.4 RUNTIME ANALYSIS

We next analyze the computational cost of our Real-to-Sim reconstruction and mass identification pipeline. For each object, we capture approximately 300–340 RGB images. The end-to-end offline reconstruction—Structure-from-Motion via COLMAP Pan et al. (2024); Schönberger and Frahm (2016) followed by 3DGS and 2DGS—typically requires 30–35 minutes per object on our device. Representative statistics are reported in Table 4. Despite mesh complexity varying from $\sim 1.3 \times 10^4$ to $\sim 6.8 \times 10^4$ vertices, reconstruction time remains within a narrow range across objects.

Mass identification via differentiable physics scales with the number of mesh vertices. On our hardware, each training iteration takes approximately 1.43–1.68 s, and convergence is typically achieved within about 200 epochs. In practice, this corresponds to 5–20 minutes for most of the objects listed in Table 4. All reconstruction and mass-identification steps are performed offline and thus do not affect the real-time execution of the learned grasping policy.

### 6 CONCLUSION

D-REX is a real-to-sim-to-real framework that leverages differentiable simulation to create visually realistic and physically accurate digital twins from visual observations and robot control signals, enabling robust dexterous grasping policies. Through identifying object mass through robot-object interactions, it achieves generalization across diverse object shapes and densities. Furthermore, integrating force-aware control conditioned on mass into imitation learning enhances policy robustness and adaptability, thus offering promising potential for scalable data generation and the development of generalizable policies, representing a significant step toward robust real-world robotic systems.

### 7 REPRODUCIBILITY STATEMENT

We rely on several open-source foundation models. The implementation details necessary to reproduce our experiments are provided in the Appendix and Supplementary Material. For our primary contributions—learning mass from video and the force–position hybrid policy—we include the codebase and dataset in the Supplementary Material and our website.

## 8 ETHICS STATEMENT

This work adheres to the ICLR Code of Ethics; all authors have read and acknowledge the Code as part of the submission process. Where our research involved human annotators or user data, we obtained approval or documented exemption from an institutional ethics review board and informed consent, used only data collected with permission, applied de-identification, and restricted access; otherwise we relied on publicly available datasets under their licenses. We assessed potential risks—including privacy, security, bias/discrimination, dual-use or misuse, and environmental impact—documenting limitations, failure modes, and mitigations to minimize harm. We also disclose any use of large language models (LLMs) in ideation, coding, or writing and accept full responsibility for all content; LLMs are not authors.

## 9 ACKNOWLEDGEMENT

The USC Physical Superintelligence Lab acknowledges generous supports from Toyota Research Institute, Dolby, Google DeepMind, Capital One, Nvidia, Bosch, NSF, and Qualcomm. Yue Wang is also supported by a Powell Research Award.

This work benefited from discussions with Professor Stephen Tu, Dr. Quankai Gao, and Dr. Yuming Gu.

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

# A APPENDIX

## A.1 PRELIMINARIES

This paper aims to accurately reconstruct the physical process of human hand grasping using only visual observations and robot control signals, without requiring access to ground-truth physical parameters. Our approach is grounded in two key components. The first is a differentiable, particle-based physics simulation engine Freeman et al. (2021), which enables gradient-based optimization of physical properties such as object mass. The second is a Real-to-Sim reconstruction pipeline based on Gaussian Splatting Lou et al. (2024), which allows us to build photorealistic and spatially consistent 3D scenes from video input. By combining these two components, we construct a fully differentiable pipeline that bridges real-world perception and physical simulation, supporting accurate modeling of dynamic hand-object interactions and enabling robust policy learning in simulation.

Robotic simulation engines such as MuJoCo Todorov et al. (2012), Isaac Sim Mittal et al. (2023), and GradSim Jatavallabhula et al. (2021b); Fuji-Tsang et al. (2019) are fundamentally built upon the Lagrangian formulation of mechanics Li et al. (2023b), which models the evolution of physical systems by tracking a fixed set of particles or reference points through space and time. This approach assumes a consistent and predefined structure in the simulation environment, typically described using formats such as MJCF or URDF. These configurations specify the number and arrangement of system components, such as joints, links, and actuated elements, which remain constant throughout the simulation. At each discrete timestep, the state of every object is updated based on dynamic and kinematic equations that reflect the physical principles embedded in the simulation engine. As a result, the evolution of object poses, velocities, and contact interactions is governed by the engine's internal numerical solvers and integration schemes. This structured and physics-informed representation is crucial for accurately modeling force transmission, contact behavior, and motion in robotic grasping scenarios.

**Differentiable Physics.** A foundational assumption of our engine is that once the static scene reconstruction is completed, the physical configuration of the environment remains unchanged throughout the system identification and policy training stages. That is, no additional objects or robots are introduced to, nor are existing components removed from, either the simulation environment or the real-world scene. Consequently, the states of all entities captured during the observation phase remain consistent and are used directly for deploying control signals in both simulation and real-world execution. This guarantees the fidelity of simulation rollouts and the alignment of dynamics between domains.

Our system architecture is governed by two fundamental categories of equations. The first involves *kinematic equations* Corke (2007), which model the articulated motion of the robotic arm and hand, accounting for joint angles, velocities, and end-effector trajectories. These equations underpin the robot's ability to reach and manipulate objects in a controlled fashion. The second set comprises *dynamic equations* Jatavallabhula et al. (2021b), which govern the interactions between the object, the robotic hand, and the supporting surface (e.g., table). These dynamics describe the forces and torques that arise during contact, enabling accurate simulation of object responses.

To simulate and optimize object behavior, we employ a dual-engine architecture consisting of MJX (the JAX-based backend of Brax) and GradSim. For spatial representation within the differentiable physics engine, we use the object's mesh vertices as the fundamental particles. These vertices serve as geometric and physical descriptors that enable fine-grained modeling of object-hand and object-environment interactions.

MJX is used to model robot kinematics and extract detailed contact information during simulation rollouts. It provides precise contact points, surface normals, and force vectors arising from interactions with the robotic hand. This information is crucial for establishing accurate boundary conditions for system identification and subsequent policy learning.

In parallel, GradSim Jatavallabhula et al. (2021b) offers a PyTorch-based engine for gradient-based simulation of object dynamics. It models the effects of gravity, inertial forces, and external perturbations (such as pushes from the robot or collisions with the ground), enabling smooth gradient flow through time. This setup facilitates efficient mass parameter optimization and supports end-to-end training pipelines involving both perception and control.

A key assumption in our setup is that the relative poses between the object, the ground, and the robotic hand within the simulation closely approximate those in the real world. This alignment is critical to ensure that simulated contact events reflect real-world conditions, enabling high-fidelity modeling of physical interactions. To this end, we align object placement using estimated poses obtained from visual tracking pipelines such as FoundationPose, ensuring consistent coordinate frames.

Although our engine incorporates Position-Based Dynamics (PBD) Müller et al. (2007) for stability and efficiency, we introduce tailored modifications to enhance collision detection and contact resolution. Specifically, we refine the broad-phase collision detection algorithm to better handle high-resolution meshes and non-convex geometries. This is essential for accurately modeling complex objects with fine surface detail and for ensuring robust gradient propagation during contact-rich interactions.

By combining MJX's strengths in kinematic modeling and contact extraction with GradSim's gradient-based physical simulation, our engine enables end-to-end mass identification and force-aware policy training. These capabilities lay the foundation for accurate and generalizable robotic grasping in real-world settings, bridging the sim-to-real gap through physically grounded learning.

### A.1.1 PARTICLE-BASED PHYSICS SIMULATION

Particle-based physics simulation is extensively used in computational physics and graphics for modeling dynamic behaviors of objects Jiang et al. (2016). Unlike traditional methods that rely on continuous volumes or polygonal meshes, particle-based methods discretize objects into numerous discrete particles, each endowed with physical attributes such as mass $m_i$, position $\mathbf{x}_i$, and velocity $\mathbf{v}_i$, as well as material properties including elasticity, friction, and damping. This discrete representation allows the efficient and realistic simulation of complex behaviors, especially beneficial in scenarios involving deformable or fragmented objects, fluids, and granular materials.

The center of mass $\mathbf{COM}$ for a particle-based system can be computed by:

$$\mathbf{COM} = \frac{\sum_i m_i \mathbf{x}_i}{\sum_i m_i}. \tag{12}$$

The inertia tensor $\mathbf{I}$, which describes an object's resistance to rotational acceleration, is computed relative to the center of mass as:

$$\mathbf{I} = \sum_i m_i \left[ \|\mathbf{r}_i\|^2 \mathbf{E} - \mathbf{r}_i \mathbf{r}_i^\top \right], \quad \text{where} \quad \mathbf{r}_i = \mathbf{x}_i - \mathbf{C} \tag{13}$$

with $\mathbf{E}$ denoting the identity matrix.

**Position-Based Dynamics (PBD).** Position-Based Dynamics (PBD) is a widely adopted paradigm in real-time and interactive physics simulation due to its stability, simplicity, and efficiency in handling constraint-driven dynamics Bender et al. (2017). Unlike traditional force-based methods that compute motion by integrating forces and torques explicitly, PBD enforces physical consistency by iteratively projecting particle positions to satisfy a set of predefined geometric and physical constraints. This projection-based formulation naturally accommodates large simulation time steps, making it particularly suitable for high-speed applications such as robotic grasping and interactive environments.

**Prediction Step (Implicit Integration).** The simulation begins by predicting particle states using semi-implicit Euler integration, which offers numerical stability and reduces oscillations during stiff interactions. For each particle $i$, the translational motion is computed as:

$$\mathbf{v}_i^{t+\Delta t} = \mathbf{v}_i^t + \Delta t \frac{\mathbf{f}_i^t}{m_i}, \tag{14}$$

$$\mathbf{x}_i^{t+\Delta t} = \mathbf{x}_i^t + \Delta t \, \mathbf{v}_i^{t+\Delta t}, \tag{15}$$

where $\mathbf{f}_i^t$ is the external force (e.g., gravity or contact impulses), $m_i$ is the particle mass, and $\Delta t$ is the simulation timestep. For rigid-body components, angular motion is predicted using:

$$\boldsymbol{\omega}_i^{t+\Delta t} = \boldsymbol{\omega}_i^t + \Delta t \, \mathbf{I}_i^{-1} \left( \boldsymbol{\tau}_i^t - \boldsymbol{\omega}_i^t \times (\mathbf{I}_i \boldsymbol{\omega}_i^t) \right), \tag{16}$$

$$\mathbf{q}_i^{t+\Delta t} = \mathbf{q}_i^t + \frac{\Delta t}{2} \tilde{\boldsymbol{\omega}}_i^{t+\Delta t} \mathbf{q}_i^t, \tag{17}$$

where $\mathbf{I}_i$ is the inertia tensor, $\boldsymbol{\tau}_i^t$ is the external torque, and $\mathbf{q}_i^t$ is the orientation represented as a unit quaternion. Here, $\tilde{\boldsymbol{\omega}}_i = [0, \boldsymbol{\omega}_i^\top]^\top$ embeds angular velocity into the quaternion algebra.

**Constraint Projection Step.** Once predicted states are available, positional constraints are enforced through iterative corrections. Each constraint $C(\mathbf{x}_i, \mathbf{q}_i) \geq 0$ represents a physical requirement (e.g., no interpenetration, fixed distances, volume preservation) and is resolved using a gradient-based position correction scheme. For constraint satisfaction, the positional update is computed as:

$$\Delta\mathbf{x}_i = -\lambda \frac{1}{m_i} \nabla_{\mathbf{x}_i} C(\mathbf{x}_i), \quad \text{with} \quad \lambda = \frac{C(\mathbf{x}_i)}{\sum_j \frac{1}{m_j} \|\nabla_{\mathbf{x}_j} C(\mathbf{x}_j)\|^2}, \tag{18}$$

where the Lagrange multiplier $\lambda$ ensures physically consistent constraint enforcement. Iterative Gauss-Seidel or Jacobi solvers are used to converge the system to a valid constraint-satisfying configuration.

**Velocity Update Step.** After the constraints are enforced, particle velocities are updated to reflect the corrected positions:

$$\mathbf{v}_i^{t+\Delta t} \leftarrow \frac{\mathbf{x}_i^{t+\Delta t} - \mathbf{x}_i^t}{\Delta t}. \tag{19}$$

This ensures consistency between position corrections and subsequent dynamics, maintaining momentum while preserving the stability advantages of PBD.

**Discussion.** The particle-based formulation enables fine-grained spatial resolution and direct grasping of geometric attributes, which is particularly beneficial for simulating high-DOF robotic hands interacting with rigid, deformable or complex-shaped objects. Furthermore, the implicit treatment of constraints circumvents many of the numerical instabilities associated with stiff force-based models, making PBD highly suitable for differentiable simulation settings where robustness and gradient flow are important Standley et al. (2017).

### A.1.2 GAUSSIAN SPLATTING

Gaussian Splatting has emerged as a powerful technique in robotic real-to-sim pipelines for capturing scenes, objects, and backgrounds with high geometric fidelity and photorealistic detail. It enables flexible and efficient modeling of complex environments from monocular video input, facilitating accurate spatial reconstruction and rendering. In our engine, we adopt the real-to-sim pipeline proposed in Lou et al. (2024), which transforms real-world scanned videos into simulation-ready assets. By leveraging Gaussian Splatting, we efficiently align the reconstructed object meshes with the simulation environment, enabling seamless integration.

To further enhance geometric consistency, we incorporate the stable normal constraint introduced in Ye et al. (2024b;a), which enforces consistent surface normals across reconstructed points. This constraint is particularly important for preserving fine surface details and mitigating noise, especially in scenes with complex geometry or intricate textures.

Together, this process allows us to recover two critical components for our differentiable physics modeling: (1) the object's 3D geometry and (2) its relative pose with respect to the robotic arm, both of which are essential for accurate system identification and simulation alignment.

### A.2 IMPLEMENTATION DETAILS

### A.2.1 IMPLEMENTATION OF REAL-TO-SIM RECONSTRUCTION

We begin by constructing a visually and geometrically precise digital twin of the target environment, leveraging a particle-based Gaussian splatting approach Kerbl et al. (2023); Huang et al. (2024). From environment-centric ($\mathcal{I}_s$) video streams captured by a mobile device, we obtain calibrated camera trajectories via structure-from-motion (SfM) Pan et al. (2024); Schönberger and Frahm (2016). The pipeline then trains two disjoint ensembles of Gaussian primitives, each pursuing a separate objective.

**1) Volumetric rendering set.** We maintain a set of 3D Gaussians

$$\mathcal{P}^{\text{rend}} = \left\{ (x_i, y_i, z_i, r_i, g_i, b_i, o_i, s_i, \Sigma_i) \right\}_{i=1}^{N_{\text{rend}}},$$

where $(x_i, y_i, z_i) \in \mathbb{R}^3$ is the center of the $i$-th Gaussian, $(r_i, g_i, b_i) \in [0, 1]^3$ its RGB color, $o_i \in [0, 1]$ the opacity coefficient for alpha blending, $\Sigma_i \in \mathbb{R}^{3 \times 3}$ a symmetric positive-definite covariance specifying anisotropic extent, $s_i$ represent the semantic and instance id of the gaussian, and $N_{\text{rend}}$ the total count of such primitives. These particles are optimized exclusively for photometric fidelity, enabling differentiable volume splatting and achieving real-time novel-view synthesis.

**2) Surface reconstruction set.** Geometry is approximated with a separate set of 2D surface-aligned Gaussians

$$\mathcal{P}^{\text{surf}} = \left\{ (x_j, y_j, z_j, \mathbf{t}_{u,j}, \mathbf{t}_{v,j}, s_{u,j}, s_{v,j}) \right\}_{j=1}^{N_{\text{surf}}},$$

where $(x_j, y_j, z_j) \in \mathbb{R}^3$ represents the disk center, $\mathbf{t}_{u,j}, \mathbf{t}_{v,j} \in \mathbb{R}^3$ are orthonormal tangent vectors, and $s_{u,j}, s_{v,j} > 0$ set the standard deviations along those directions. The outward surface normal is

$$\mathbf{n}_j = \mathbf{t}_{u,j} \times \mathbf{t}_{v,j}.$$

This ensemble is trained with depth distortion and normal consistency terms for geometric accuracy, remaining untouched by photometric loss.

After training, the surface Gaussians in $\mathcal{P}^{\text{surf}}$ are rasterized into multi-view depth maps, fused into a truncated signed-distance field, and converted via marching cubes into a triangle mesh. Surface normals are estimated Ye et al. (2024a), giving the final collision mesh $\mathcal{M}$. Since $\mathcal{P}^{\text{rend}}$ and $\mathcal{P}^{\text{surf}}$ do not share parameters and employ disjoint loss functions, improvements in appearance do not degrade geometric fidelity.

### A.2.2 CONSTRUCTING MJCF MODELS USING RECONSTRUCTED GAUSSIAN AND MESH REPRESENTATIONS

The MuJoCo XML Control Format (MJCF) encodes key simulation components, including an object's kinematic structure, PID control gains, stiffness parameters, collision geometries $\mathcal{K}$ along with the surface point cloud $\mathcal{P}^{\text{surf}}$ Zeng et al. (2017), and specifications of actuated joints. To construct a complete MJCF model from our reconstructed Gaussian splats and mesh representations, we first embed the static environment as an unmovable background and define the reconstructed object as a free joint body within the simulation environment.

We then align the reconstructed Gaussian coordinate frame and chirality with MuJoCo's convention, following the transformation procedure described in Lou et al. (2024). To ensure simulation realism, we extract the relative pose between the object and the robotic arm in the real-world scene and apply this transformation as the initial configuration of the free joint object in simulation. After integrating all relevant positional and control information, we use Vision-Language Models (VLMs) to infer initial estimates of physical parameters, including object mass, which are critical for downstream simulation fidelity.

The resulting MJCF model, with accurately aligned coordinates, initial pose, and geometry, provides a strong foundation for subsequent system identification and physics-based policy learning. It also enables high-fidelity rendering and precise real-to-sim transitions.

### A.2.3 IMPLEMENTATION OF MASS IDENTIFICATION

This section addresses two key aspects of our mass identification engine: (1) the strategy for mass-inertia modeling, and (2) the set of adaptive parameters necessary to support mass learning across objects with diverse physical properties and geometric variations.

**Mass-Inertia Modeling.** In conventional settings, an object's ground-truth mass is typically distributed uniformly across its constituent particles, as defined in Equation 12. However, this strategy often leads to numerical instability and gradient explosion within real-to-sim-to-real optimization engines, particularly when dealing with high-resolution objects that contain over 50,000 vertices but possess relatively low mass Chen et al. (2025b). Under such conditions, the resulting average particle mass can fall below $10^{-6}$ kg, introducing significant numerical errors.

To mitigate this issue, we assign the full object mass to each particle. Gravitational forces are uniformly applied to all particles, and external forces are scaled proportionally to the number of sampled vertices. This formulation preserves numerical stability by avoiding exceedingly small per-particle mass values.

Additionally, because the number of vertices varies across reconstructed objects, we adaptively select a subset of active vertices that lie on contact surfaces between the object and the robotic fingers. This further improves simulation fidelity and ensures relevant physical interactions are emphasized.

To guarantee consistency between the real-world observations and simulation environment, we explicitly synchronize frame rates, temporal bounds (start and end times), and spatial centering between the FoundationPose tracking system and the MuJoCo simulation defined in MJCF format.

**Contact Modeling, Explicit Gradient Representation, Adaptive Learning Parameters.** We extract contact points and corresponding forces from robotic action rollouts conducted in both simulated and real-world environments. In the simulation, following the real-to-sim reconstruction, objects are placed in relative positions consistent with their real-world configurations. To ensure stable contact modeling within a Position-Based Dynamics (PBD) engine, objects are initialized slightly above the ground (e.g., $[0.05, 0.05, \frac{\text{Height}}{2} + 0.01]$), preventing premature ground contact and maintaining simulation stability.

Precise temporal synchronization across real-world object trajectories, robot control signals, and simulation rollouts is essential for reliable mass identification. We leverage FoundationPose Wen et al. (2024) to obtain accurate object pose estimates, and align simulation timelines accordingly to ensure consistency between observed and simulated motion.

For explicit gradient computation, we implement a semi-implicit integration scheme following the formulation introduced in Jatavallabhula et al. (2021a), enabling differentiable backpropagation through contact events and object dynamics.

**Adaptive Learning Strategy** To accommodate objects with varying mass scales, we employ an adaptive learning strategy. Initially, particle masses are uniformly set to approximately 0.002 kg per vertex, but this baseline must be adjusted according to the object's overall mass to ensure stable convergence. For heavier objects, such as a ketchup bottle (0.8 kg), training requires higher learning rates and longer schedules, often up to 2000 epochs, to achieve convergence. In contrast, medium-mass objects (0.1 kg) typically converge efficiently within 100 epochs using a moderate learning rate. Lightweight objects (0.05 kg) benefit from learning rate decay and similarly converge within 100 epochs.

Successful mass learning also depends on several key factors. The duration of the applied impulse, determined by the active contact interval between the robotic fingers and the object, directly influences the estimated dynamics. We select the active tracking frame from FoundationPose to mark the critical transition from motion onset to rest. Additionally, we apply a canonical re-centering vector to align object positions in simulation space, reducing variation introduced by camera viewpoint differences. Finally, the estimated contact area is adjusted proportionally to the object's vertex count and active contact regions, allowing accurate modeling of the hand-object interaction Lu et al. (2023); Bronstein et al. (2022).

**Discussion of the $z$-axis error from FoundationPose.** In practice, the $z$-axis error produced by FoundationPose is non-negligible. We observe that, in some cases, this error must be manually post-processed to better align the estimated pose with the real-to-sim digital asset.

### A.2.4 ABLATION STUDY ON SEMI-IMPLICIT EULER AND EXPLICIT EULER

In this supplementary section, we expand the parameter-identification formulation and provide a more formal ODE-based description of our objective, including boundary and initial conditions, sampling, and convergence properties.

The object dynamics in our setup are modeled by the Newton–Euler equations

$$m\ddot{x}(t) = f_{\text{contact}}(x(t), u(t)) + f_{\text{grav}}, \tag{20}$$

where $x(t)$ denotes the object pose (or the position of its center of mass), $u(t)$ encodes the robot hand actions, and $m$ is the unknown object mass.

The initial conditions and geometric constraints are determined by: (i) the object geometry and pose reconstructed via Gaussian splatting (initial pose/shape), (ii) the robotic hand configuration and

known actuation commands (control input $u(t)$), and (iii) the ground plane and environment, also provided by the reconstruction.

The dynamics residual is evaluated at discrete time steps $\{t_k\}_{k=1}^{T}$ along the observed trajectory, with contact forces computed from the mesh vertices of the Gaussian-splatting reconstruction. The ground-truth trajectories $\{\hat{x}_k\}_{k=1}^{T}$ at times $\{t_k\}_{k=1}^{T}$ are provided by FoundationPose. Our differentiable simulator enforces the governing Newton–Euler equations at these sampled points, so that convergence of the learned dynamics and mass parameter is driven by minimizing the residual of a well-posed ODE system.

In the specific "push-down" interaction used for mass identification, let $z(t)$ denote the object position along the push axis, and $u(t)$ the known contact force applied by the robot:

$$m\ddot{z}(t) = u(t) - mg. \tag{21}$$

Integrating twice in time, we obtain

$$z(t;m) = z_0 + v_0 t + \frac{1}{m}\int_0^t \int_0^\tau u(s)\, ds\, d\tau - \frac{1}{2}gt^2, \tag{22}$$

which can be written as

$$z(t;m) = \alpha(t) + \frac{1}{m}\,\beta(t), \tag{23}$$

where $\alpha(t)$ and $\beta(t)$ depend only on known quantities (initial conditions and measured robot forces). Thus, under this simplified contact model, the trajectory is *affine in the inverse mass* $1/m$. Discretizing time gives

$$z_k(m) = \alpha_k + \frac{1}{m}\,\beta_k, \quad k = 1,\ldots,T. \tag{24}$$

The trajectory loss we use,

$$\mathcal{L}_{\text{traj}}(m) = \sum_{k=1}^{T} \left\| z_k(m) - \hat{z}_k \right\|^2, \tag{25}$$

then becomes a standard least-squares problem in the parameter $\theta = 1/m$. In this parametrization, $\mathcal{L}_{\text{traj}}(\theta)$ is a quadratic function of $\theta$ and therefore convex. In practice, our learning procedure operates through vector–Jacobian products in the differentiable simulator, but the underlying structure is equivalent to solving a well-conditioned least-squares identification problem for the mass.

We now clarify the effect of using a semi-implicit Euler scheme versus an explicit Euler scheme in our differentiable simulator.

For the discrete-time dynamics, we adopt a semi-implicit (symplectic) Euler scheme. For a time step $\Delta t$, the updates are

$$v_{k+1} = v_k + \Delta t\, m^{-1} f(x_k, u_k), \tag{26}$$
$$x_{k+1} = x_k + \Delta t\, v_{k+1}, \tag{27}$$

where $f(x_k, u_k)$ includes gravity and contact forces. In contrast, the explicit Euler scheme would use

$$v_{k+1} = v_k + \Delta t\, m^{-1} f(x_k, u_k), \tag{28}$$
$$x_{k+1} = x_k + \Delta t\, v_k. \tag{29}$$

This semi-implicit update rule is closely related to the integrators commonly used in position-based dynamics (PBD) methods, and is known to have a larger stability region for stiff systems.

In contact-rich dynamics, especially for small masses or stiff contact forces, explicit Euler has a much smaller stability region and tends to produce large, oscillatory updates in $x_k$. This is particularly problematic in our setting because we initialize the mass at a relatively small value (a conservative lower bound), which amplifies the effective acceleration $m^{-1}f$ and can lead to very large position updates and exploding gradients when using explicit Euler. Semi-implicit Euler, on the other hand, updates the velocity first and then uses the updated velocity to advance the position, which empirically leads to more stable and accurate trajectories.

We empirically compare the two schemes on our push-based mass-identification tasks. Table 5 reports both the identified mass (in grams) and the average optimization time per iteration (in seconds) for each object:

| Object | Ground truth (g) | Semi-implicit (g) | Explicit (g) | Semi-implicit (s) | Explicit (s) |
|---|---|---|---|---|---|
| Lego | 59 | 51 | 34 | 1.38 | 1.21 |
| Ketchup | 726 | 667 | 685 | 1.43 | 1.19 |
| Cookie | 210 | 200 | 189 | 1.41 | 1.18 |
| Domino | 106 | 117 | 135 | 1.36 | 1.17 |
| U | 125 | 110 | 98 | 1.39 | 1.20 |
| A | 134 | 145 | 120 | 1.42 | 1.22 |

Table 5: Ablation study comparing semi-implicit and explicit Euler schemes for mass identification. We report the ground-truth mass, the identified mass under each integrator, and the average optimization time per iteration (in seconds). Semi-implicit Euler yields more accurate mass estimates at a modest runtime cost.

---

**Algorithm 1** Force-Aware Policy Training

---

**Input:** Set of object meshes and masses: $\{(\mathbf{K}_i, \mathbf{M}_i)\}_{i=1}^N$
**Output:** Learned actions and forces: $\{(\mathbf{Action}_i, \mathbf{Force}_i)\}_{i=1}^N$

1: **for** each demonstration $(\mathbf{K}_i, \mathbf{M}_i)$ **do**
2:     Extract human hand poses and object poses using HaMeR Pavlakos et al. (2024) and MCC-HO Wu et al. (2024a).
3:     Retarget human hand poses and corresponding end-effector poses onto the robotic hand.
4:     **Positional Encoding:** Encode vertices using positional encoding to obtain feature representations.
5:     **Dataset Construction:** Prepare training batches comprising encoded vertices, object mass $\mathbf{M}_i$, and ground-truth actions. Load corresponding MJCF files generated by Real2Sim.
6:     **Stage One Training (Supervised):** Train the policy network by setting force and contact head ground-truth labels to $1$, optimizing initial grasp prediction.
7:     **Stage Two Training (Simulation-based Refinement):** Roll out predicted actions within the MuJoCo simulator using the Real2Sim-generated MJCF files. Compute force and contact rewards from simulation outcomes and perform backpropagation to refine the model.
8:     **Real-world Deployment:** Deploy the grasping policy onto the real robotic system using the reconstructed object mesh, executing predicted actions with force control.
9: **end for**

---

Overall, explicit Euler is slightly faster per iteration but consistently less accurate, with larger deviations from the ground-truth mass and more frequent numerical instabilities during optimization. Semi-implicit Euler provides a better trade-off between stability and accuracy, which is crucial for reliable differentiable mass identification in our Real2Sim pipeline.

### A.2.5 IMPLEMENTATION OF D-REX'S GRASPING POLICY

Table 6 details the neural network architecture used in our *GraspMLP*, while Algorithm 1 and Algorithm 2 describe the training pipeline. For standard objects, the grasping policy is trained with approximately 200 demonstrations per object. For objects with higher geometric or dynamic complexity, we scale the dataset to include up to 5000 demonstrations, ensuring sufficient coverage of the variance necessary for robust policy learning. Empirically, we find that the integration of a lightweight policy network, accurate modeling of human hand-object interactions, and precise physics-informed constraints enables reliable and high-performance grasping behavior tailored to each object.

### A.2.6 IMPLEMENTATION DETAILS OF THE FORCE-BASED GRASPING POLICY

For the grasp-position policy, we use a two-stage training procedure that combines human demonstrations with simulation-based refinement (see Appendix A.2.4 for additional details). Our grasping policy, denoted *GraspMLP*, is first trained on human data and subsequently refined in simulation.

**Input representation and supervision from human demonstrations.** We begin by collecting human grasp demonstrations and reconstructing the human hand and object poses using

---

**Algorithm 2** Two-phase Training Procedure

---

1: **Initialize:** model parameters $\theta$, optimizer, dataloader $\mathcal{D}$, environment $\mathcal{E}$, loss functions: MSELoss ($\mathcal{L}_{\text{MSE}}$), BCELoss ($\mathcal{L}_{\text{BCE}}$).
2: **Phase 1: Supervised Pre-training**
3: **for** epoch $= 1, \ldots, E_1$ **do**
4:     **for** batch $(x, a, r, f) \sim \mathcal{D}$ **do**
5:         Compute predictions: $(\hat{a}, \hat{r}, \hat{f}) \leftarrow \text{model}(x; \theta)$
6:         Compute losses:
7:           $\mathcal{L}_a \leftarrow \mathcal{L}_{\text{MSE}}(\hat{a}, a)$
8:           $\mathcal{L}_r \leftarrow \mathcal{L}_{\text{BCE}}(\hat{r}, r)$
9:           $\mathcal{L}_f \leftarrow \mathcal{L}_{\text{MSE}}(\hat{f}, f)$
10:         Backpropagate total loss: $\mathcal{L} = \mathcal{L}_a + \mathcal{L}_r + \mathcal{L}_f$
11:         Update parameters $\theta$
12:     **end for**
13: **end for**
14: **Phase 2: Environment Interaction**
15: **for** epoch $= 1, \ldots, E_2$ **do**
16:     **for** batch $x \sim \mathcal{D}$ **do**
17:         Predict actions and rewards: $(\hat{a}, \hat{r}, \hat{f}) \leftarrow \text{model}(x; \theta)$
18:         Execute $\hat{a}$ in environment $\mathcal{E}$ and observe rewards $r_{\text{env}}$ and contact-based forces $f_{\text{env}}$
19:         Compute scaled ground-truth force: $f_{\text{env}} = \text{clip}(\frac{m \cdot g \cdot \text{num\_contacts}}{f_{\text{max}}}, 0, 1)$
20:         Compute losses:
21:           $\mathcal{L}_r \leftarrow \mathcal{L}_{\text{BCE}}(\hat{r}, r_{\text{env}})$
22:           $\mathcal{L}_f \leftarrow \mathcal{L}_{\text{MSE}}(\hat{f}, f_{\text{env}})$
23:         Backpropagate weighted loss: $\mathcal{L} = 0.8\mathcal{L}_r + 0.3\mathcal{L}_f$
24:         Update parameters $\theta$
25:     **end for**
26: **end for**

---

HaMeR Pavlakos et al. (2024) and MCCHO Wu et al. (2024a). These human hand trajectories are retargeted to the target robotic hand, yielding joint-space reference actions. For each object, we use the reconstructed mesh from the Real2Sim pipeline together with its identified mass. We uniformly sample mesh vertices, apply positional encodings to their 3D coordinates, and concatenate these features with the object mass to form the per-vertex input to GraspMLP.

**Network architecture.** GraspMLP consists of a shared MLP backbone that processes the mass-conditioned per-vertex features, followed by three task-specific heads: (i) a 16D head for joint-space grasp actions, (ii) a 2D head that predicts a contact-based reward/feasibility score, and (iii) a 1D head that outputs a normalized grasp force in $[0, 1]$. The full architecture (layer widths and activations) is summarized in Table 4 of the main paper.

**Phase 1: Supervised pre-training.** In the first phase, we train GraspMLP on human demonstrations to regress joint actions and predict contact/force labels. We use an $\ell_2$ (MSE) loss for the joint actions and the normalized force output, and a binary cross-entropy (BCE) loss for the contact/feasibility head (Algorithm 2, lines 3–13). All demonstrated grasps are treated as successful in this stage, and the contact/force labels are derived directly from the human data.

**Phase 2: Simulation-based refinement.** In the second phase, we refine the policy in simulation. We roll out GraspMLP in MuJoCo using the Real2Sim-generated MJCF models and the identified object masses (Algorithm 1, lines 6–7; Algorithm 2, lines 15–26 in the Appendix). From these rollouts, we compute: (i) a contact-based reward that reflects grasp success and stability, and (ii) a scaled force target based on object mass, gravity, and the number of active contact points. We then update GraspMLP with a weighted combination of BCE loss (for the reward/feasibility signal) and MSE loss (for the force targets), effectively performing RL-style fine-tuning while remaining in a supervised learning framework.

**Deployment and force control.** At deployment time, the policy takes as input the observed object point cloud (in the same format as the training vertices) and the identified mass. On real hardware, we implement a force-based controller by mapping the predicted normalized grasp force to actuator currents. For Allegro and LEAP hands, which employ direct-drive brushless motors, the mapping from motor current to fingertip force is approximately linear, so current control serves as a reliable proxy for controlling grasp force.

**Applicability to different object scales.** Although our experiments focus on objects of standard tabletop scale, our method is not intrinsically limited to these sizes. Both the mass-identification and grasping modules operate on pose trajectories and reconstructed geometry, and thus can in principle be applied to smaller objects as well. In practice, however, very small objects are constrained by two factors: (i) the perception pipeline must still provide accurate 6-DoF poses and sufficiently detailed geometry, and (ii) the robot hardware (including end-effector design, finger size, and sensing resolution) must be capable of reliably grasping and manipulating such objects. In our current setup, these perception and hardware constraints are the main bottlenecks for tiny objects, rather than limitations of the Real2Sim or mass-identification formulation itself. These challenges are shared by most pose-based manipulation methods. Consequently, in this work we focus on object sizes that are standard in existing 6-DoF pose estimation benchmarks and robotic grasping setups, where both perception and hardware are reliable.

### A.2.7 COMPUTATIONAL DETAILS AND TIMINGS

Our grasping policy is trained on datasets containing 200 to 300 demonstration poses per object by default, which results in a training duration of approximately 2 minutes per object on one NVIDIA RTX 4090 GPU. For more complex or high-variance objects that require additional data coverage, we scale the training dataset to include up to 5000 demonstrations. In such cases, the training time increases to approximately 20 minutes per object, due to the additional dataset batch size.

Inference is highly efficient. Once the policy is trained and deployed, it requires only a reconstructed URDF or MJCF representation as input, capturing the object's geometry, pose, and physical properties. Given such input, the policy predicts a stable grasp configuration in approximately 0.5 seconds per object pose. This low-latency inference time makes the system practical for real-time and on-robot applications, particularly in scenarios that demand quick adaptation to dynamic object placements or orientations.

Overall, our engine demonstrates a favorable trade-off between training cost and deployment efficiency, with scalable training capabilities and low runtime overhead for inference.

| Component | Operation | Output Dim. | Details |
|---|---|---|---|
| Input | Positional Encoding | $N \times 3$ | $N$ object vertices (XYZ) |
| Linear Layer 1 | Fully Connected | 256 | Input: $3 \rightarrow 256$ |
| Activation 1 | ReLU | 256 | Non-linearity |
| Linear Layer 2 | Fully Connected | 256 | $256 \rightarrow 256$ |
| Activation 2 | ReLU | 256 | Non-linearity |
| Linear Layer 3 | Fully Connected | 256 | $256 \rightarrow 256$ |
| Activation 3 | ReLU | 256 | Non-linearity |
| Action Head | Linear | 16 | Joint action output |
| Reward Head | Linear + Sigmoid | 2 | Contact constraint prediction |
| Force Head | Linear + Sigmoid | 1 | Grasping force prediction |

Table 6: Architecture of the proposed GraspMLP network. The input consists of per-vertex 3D coordinates. The shared backbone maps the input into a latent feature space, which is subsequently decoded into separate heads for predicting joint actions, contact-based reward signals, and grasping force.

### A.2.8 IMPLEMENTATION OF BASELINES ON OBJECT GRASPING

**Human2Sim2Robot Baseline.** In the Human2Sim2Robot engine Lum et al. (2025), we operate under the assumption that the grasping end-effector pose extracted from human demonstration videos

is both accurate and physically feasible for robot execution. These grasp poses—typically obtained from hand-object interaction sequences—are directly retargeted to the Leap Hand using the official retargeting implementation provided by the Leap Hand repository, preserving the spatial fidelity of the original grasp intent.

For a fair and consistent baseline comparison, we replace the original demonstration assets and object meshes used in Human2Sim2Robot with our own Real-to-Sim reconstructed meshes, which incorporate photogeometric fidelity and physical realism as described in Section Real2sim. Using these assets, grasping policies are trained until convergence, which generally requires approximately 20,000 training epochs to stabilize reward signals and behavior.

At deployment, we assume that the relative end-effector pose remains feasible under the Franka arm and CuRobo motion planning stack. That is, we expect the grasp pose transferred from human demonstrations to be executable without requiring additional replanning or corrections during real-world trials. While this assumption aligns with the original baseline setting, it introduces potential limitations in robustness, particularly under challenging object configurations.

It is important to note that the original controller, fabric, used in Human2Sim2Robot—including closed-loop visual servoing and grasp adjustment mechanisms—is not publicly available. Consequently, our reimplementation focuses solely on static inference: given a fixed RGBD frame and known object pose, the system predicts a single-step grasp action without online feedback or corrective replanning. This constraint is taken into account in our evaluations to ensure fair comparison.

**DexGraspNet 2.0 Baseline.** We adopt the two-stage grasping pipeline proposed in DexGraspNet 2.0 Zhang et al. (2024c), which separates grasp pose generation from execution via motion planning. However, rather than directly regressing relative translations and rotations from synthetic training data, our method infers these grasp parameters through MCC-HO, a pretrained model that extracts meaningful grasp features from real human hand-object interactions captured in video. These interactions are grounded in geometry reconstructed through our Real-to-Sim pipeline, where object vertices derived from point clouds are directly used to estimate feasible grasp poses in 3D space.

Once grasp poses are generated, we utilize CuRobo for trajectory planning and execution. The planned trajectories are constrained by the robotic arm's kinematic and dynamic limits, ensuring safe and feasible real-world deployment of the inferred grasp poses.

To ensure fair comparison with DexGraspNet 2.0, which assumes a fixed object mass of approximately 0.1 kg across all test scenarios. We limit our evaluation to objects of similar mass to match the conditions under which their policy was trained. However, in contrast to this fixed-mass assumption, our approach explicitly optimizes grasp strategies using the mass identified through our differentiable real-to-sim-to-real pipeline. This enables force-aware grasping, as the identified mass is used to refine force predictions and enhance grasp stability.

By leveraging human demonstrations and accurate physical modeling, our approach generalizes more robustly across varying object shapes and dynamic properties, offering improved realism and adaptability compared to methods relying solely on simulated training data and heuristic mass assumptions.

**Object Tracking and Motion Planning.** We employ FoundationPose Wen et al. (2024) for real-time 6-DoF object pose estimation during grasping. This robust visual tracking system provides temporally consistent pose predictions that enable dynamic, collision-aware trajectory planning for the robotic end-effector. These object pose estimates serve as a foundation for constructing grasping trajectories in cluttered or dynamic environments.

Once the object pose is reliably tracked, we incorporate wrist pose predictions generated by MCC-HO Wu et al. (2024a), a pretrained model designed to reconstruct hand-object interaction trajectories from human demonstration videos. The wrist poses extracted from these interactions represent feasible, human-derived grasping configurations. Together with the Real-to-Sim object pose, they define the target end-effector pose required for grasp execution.

To generate collision-free motion plans, we formulate a constrained inverse kinematics (IK) optimization problem using CuRobo Sundaralingam et al. (2023). Specifically, we seek the robot joint configuration $\mathbf{A}^*$ that minimizes the distance between the robot's forward kinematics (FK) output and

the desired end-effector pose $\mathbf{X}_{\text{ee}}^{\text{des}}$, while remaining within the robot's collision-free configuration space $\mathcal{Q}_{\text{free}}$:

$$\mathbf{A}^* = \arg \min_{\mathbf{A} \in \mathcal{Q}_{\text{free}}} \left\| \text{FK}(\mathbf{A}) - \mathbf{X}_{\text{ee}}^{\text{des}} \right\|_p. \tag{30}$$

Here, $\mathbf{X}_{\text{ee}}^{\text{des}}$ is derived from aligning the object pose (reconstructed via Gaussian Splatting and photogrammetry) with the wrist pose from human demonstration, forming a grounded and physically meaningful grasp target. The norm $\| \cdot \|_p$ (typically $L_2$) measures the spatial error in SE(3) between the planned and desired poses.

This enables physically plausible and task-relevant grasp execution that leverages real-world perception, human demonstration priors, and differentiable simulation to close the sim-to-real loop.

### A.2.9 REAL-WORLD EXPERIMENTS

In our experimental setup, the scene is composed of five primary components: a static table, a fixed background, a target object, a robotic arm, and a robotic hand. Both the table and background remain stationary and unchanging throughout the duration of each experiment, providing a consistent spatial context. The robotic arm and hand are fully actuated and precisely controlled, with all joint movements accurately tracked to ensure reproducibility and reliable system behavior.

The target object is entirely passive, which is not actuated or directly controlled. Its motion arises solely from physical interactions with the robotic hand, such as contact-induced forces during grasping or pushing. This object-centric dynamic behavior forms the basis for our system identification and policy learning tasks.

For visual tracking, we employ a third-person Intel RealSense D435i RGB-D camera positioned to capture the entire grasping workspace. To estimate the 6-DoF object pose over time, we use FoundationPose Wen et al. (2024), a real-time object pose estimation engine that ensures robust, frame-consistent predictions even under occlusion or clutter.

To reconstruct the geometric details of the experimental scene—including the table, object, and robot—we supplement the depth camera data with smartphone-based photogrammetry. Capturing a short monocular video using a mobile phone, we apply multi-view stereo techniques to generate dense 3D reconstructions of the environment. This process enables us to build high-resolution object meshes and spatially aligned scene representations, which are later used for initializing simulation environments and real-to-sim transfers.

Together, this combination of accurate tracking and high-fidelity geometric reconstruction provides the foundation for grounded simulation, physical parameter identification, and robust real-world policy deployment.

### A.2.10 HARDWARE SETUP

We employ two distinct robotic hands in our experimental engine to accommodate the varying requirements of system identification and dexterous grasping: the Allegro Hand and the LEAP Hand, each equipped with 16 independently actuated degrees of freedom (DoF). These platforms are selected to balance mechanical precision and torque capabilities across the experimental tasks.

Both Allegro and Leap hand are direct drive hand, which we have test our method's success rate increase. For the linkage driven hand like Inspire, we do not know the effectiveness of our force based control.

The **Allegro Hand** is a widely used 16-DoF anthropomorphic robotic hand developed specifically for research in dexterous grasping. It features internalized wiring and a compact mechanical structure, minimizing external interference during physical interactions. Its low-profile design and clean joint layout simplify kinematic and dynamic modeling, making it well-suited for physical parameter identification tasks such as object mass estimation. The reduced presence of external cabling allows for more stable contact modeling and cleaner gradient flow during differentiable physics-based optimization.

The **LEAP Hand** Shaw et al. (2023b) is a high-torque, cost-efficient robotic hand designed with modularity and real-world applicability in mind. It is constructed from a combination of 3D-printed components and off-the-shelf actuators, enabling easy customization, repair, and experimentation. Critical mechanical attributes—including finger length, joint stiffness, and inter-finger spacing—can be modified to suit specific grasping scenarios or object geometries. The LEAP Hand features a novel tendon-driven kinematic structure that enables highly dexterous and human-like articulation. Each joint is capable of exerting torques that exceed those of the human hand, while maintaining realistic velocities up to approximately 8 radians per second.

A core design principle of the LEAP Hand is to maximize the proportion of mass allocated to actuators relative to the hand's total weight, thereby enhancing grip strength while preserving a compact form factor. This focus enables it to handle heavy or irregularly shaped objects that require strong and adaptive force control. Importantly, the LEAP Hand includes integrated current- and torque-limiting mechanisms, allowing for both powerful and delicate grasping. These features make it especially suitable for executing real-world grasping tasks, where force control must be both robust and compliant.

In our experiments, we regulate the grasping force exerted by the LEAP Hand by tuning its actuator current limits, which are linearly correlated with the applied joint torques. This control scheme enables precise modulation of contact force based on object mass and surface properties, a critical requirement for sim-to-real generalization in force-aware policy learning.

By leveraging the complementary strengths of the Allegro and LEAP Hands, our engine supports both accurate physical modeling and high-performance real-world grasping, facilitating end-to-end real-to-sim-to-real learning and deployment.

**Rationale for Using Different Hands**    We employ the **Allegro Hand** for mass identification experiments due to its compact, self-contained mechanical design, which minimizes external interference. Its internalized wiring and low-torque actuation contribute to stable and noise-free contact dynamics, making it ideal for tasks that require accurate gradient propagation and precise system identification. These attributes are particularly advantageous when using differentiable physics to estimate object mass from robot-object interactions, where mechanical noise or inconsistent contact can significantly degrade optimization performance. The consistent kinematics and low-inertia structure of the Allegro Hand further improve the fidelity of object dynamics modeling during the real-to-sim identification stage.

We utilize the **LEAP Hand** Shaw et al. (2023b) for grasping and grasping tasks due to its high-torque capabilities and modular, human-like kinematic structure. The LEAP Hand features tendon-driven actuation with robust motors that can generate significantly higher forces than the Allegro Hand, enabling it to perform reliable grasps on objects with varying shapes, weights, and compliance. This is particularly important when evaluating real-world policy deployment, where robustness and grasp stability are critical. Its design prioritizes strength and dexterity, making it suitable for executing force-aware policies under physically realistic conditions. The hand's current-controlled actuation also enables precise regulation of grasping force, which we leverage in our policy to adapt to different object masses.

However, the LEAP Hand includes exposed wiring and tendon routing, which introduce mechanical noise and modeling complexity, especially during sensitive parameter estimation stages such as mass identification. These structural factors can interfere with accurate contact modeling and introduce inconsistencies in force feedback during differentiable simulation.

Through decoupling the roles of the two hands—using the Allegro Hand for precise physical parameter estimation and the LEAP Hand for robust grasping—we are able to optimize each stage of our real-to-sim-to-real engine. This separation of concerns allows our engine to balance accuracy and practicality, supporting both high-fidelity modeling and real-world deployment across a diverse set of grasping scenarios.

### A.2.11    DATASET COLLECTION AND EXPERIMENT DEPLOYMENT

To support accurate real-to-sim modeling, we collect approximately 300 RGB images per scene using a third-person RGB-D camera (Intel RealSense D435i) or scanning device like Iphone. These images are used for high-fidelity 3D reconstruction, which captures both the object geometry and

environmental context. The full reconstruction process typically takes around 30 minutes per scene and produces a Gaussian Splats representation.

Then we convert the reconstructed visual assets into simulation-ready MJCF. This conversion encodes the object geometry as collision meshes, specifies object kinematics, and initializes physical parameters for use in simulation environments such as MuJoCo. We also extract the relative pose between the object and the robotic base, which is crucial for alignment during simulation deployment.

Our experimental environment comprises a 7-DoF robotic arm (Franka Emika Panda), a dexterous robotic hand (Allegro or LEAP, depending on the task), and a static table on which the object is placed. During data collection and evaluation, the robotic system executes predefined control trajectories or learned policies while interacting with the object. Simultaneously, FoundationPose Wen et al. (2024) provides real-time 6-DoF object pose tracking using third-view RGB-D video input. This ensures precise alignment between real-world motion and the corresponding simulated trajectories.

All collected sensor data—including RGB frames, depth maps, robot joint states, and object poses—are synchronized and logged for later use in simulation, policy training, and evaluation. This structured dataset serves as the basis for mass identification and grasping policy learning, enabling consistent real-to-sim-to-real transfer across experiments.

### A.3    ABLATION STUDY

#### A.3.1    SCALING PERFORMANCE

We investigate how the number of human demonstrations influences grasping performance on a challenging object: a compact, high-density screwdriver. This object is particularly difficult to manipulate due to its small contact area and high moment of inertia, making it an ideal benchmark for evaluating the scalability of our learning engine. As illustrated in Figure 9a, we assess grasping success over 20 real-world trials using policies trained on varying numbers of demonstrations. These demonstrations are automatically filtered and converted into robot-executable grasp poses using our Real-to-Sim pipeline.

The training dataset size ranges from 255 to 6,386 grasp poses, extracted from 1 to 40 unique human video demonstrations. Our results show a clear positive correlation between demonstration count and grasping success rate: with just a handful of examples, the policy struggles to generalize and frequently fails to stabilize the object. However, as the number of demonstrations increases, the policy gains sufficient exposure to diverse object configurations and interaction patterns, enabling more robust and consistent grasps. Figure 9b provides qualitative visualizations of the grasp poses learned at different data scales. With minimal data, the policy produces suboptimal or unstable grasps, often misaligned with the object's geometry or balance point. As the dataset grows, the learned poses become progressively more aligned with physically stable and human-like strategies. These results underscore the importance of dataset scale in training force-aware grasping policies and highlight the effectiveness of our system in leveraging human video demonstrations to improve dexterous grasping performance.

#### A.3.2    ROBUST MASS IDENTIFICATION ACROSS DIVERSE OBJECTS USING DIFFERENTIABLE OPTIMIZATION

We present three representative examples of our differentiable mass optimization process in Figure 11, illustrating its convergence behavior across a diverse set of objects: Cookie, Lego, and Ketchup. In all cases, the optimization begins from a deliberately underestimated initial mass of $2\,\text{g}$—approximately $100\times$, $350\times$, and $30\times$ smaller than the ground-truth masses for the Cookie, Lego, and Ketchup, respectively.

For the Cookie object, which has moderate mass and contact dynamics, the optimization converges smoothly to the correct value despite the large initial gap. This demonstrates the robustness of our engine under mild mass discrepancies. In the case of the Lego object, which features a small contact surface and lower inertia, the large initial error induces an early overshoot. Nonetheless, the gradient-based optimizer is able to recover and guide the system toward the correct mass value within a stable number of iterations. The Ketchup bottle presents the most challenging case due to its high mass and complex geometry. The significant mismatch between the initial and true mass results in

| Method | Opt. runtime | PSNR ↑ | SSIM ↑ | LPIPS ↓ |
|---|---|---|---|---|
| 4DGS Wu et al. (2024b) | 30 min | 17.14 | 0.632 | 0.411 |
| D-REX (3DGS + opt. traj.) | 20 min | **23.54** | **0.783** | **0.237** |

Table 7: Real-to-sim reconstruction quality on the real push–ketchup sequence. Our D-REX pipeline improves photometric metrics over vanilla 4DGS while reducing optimization time.

a high initial loss. However, by applying an adaptive learning rate and increasing the number of training epochs, the system successfully converges to an accurate mass estimate.

These examples collectively highlight the flexibility and effectiveness of our differentiable engine. Regardless of the object's scale or dynamic properties, our method reliably refines mass estimates from poor initializations, enabling physically grounded simulation essential for force-aware policy learning.

### A.3.3    REAL-TO-SIM RECONSTRUCTION QUALITY

We first evaluate the quality of the reconstructed 4D scenes used for mass identification and policy learning. For the real push–ketchup sequence in Sec. 5.1, we render the reconstructed scene along the executed robot trajectory and compare it against the recorded camera frames. A video of this sequence, as well as an interactive 3DGS viewer, are provided on our anonymous project website (sections "Mass Identification via Object Pushing" and "Interactive 3D Workspace"). The reconstructions closely reproduce the observed object motion and contact interactions, indicating that the Real-to-Sim mapping is visually well aligned with the real-world data.

To quantitatively assess Real-to-Sim fidelity, we compare our D-REX Real2Sim pipeline (3DGS + optimized trajectory) with 4D Gaussian Splatting (4DGS) Wu et al. (2024b) on the same push–ketchup sequence. Following the standard D-NeRF bouncing-ball protocol, we evaluate rendering quality along the robot trajectory using PSNR, SSIM, and LPIPS, and we also report optimization runtime. Results are summarized in Table 7.

D-REX clearly outperforms vanilla 4DGS on all three photometric metrics while requiring less optimization time. We attribute these gains to supervising directly on the 6-DoF object pose and modeling the manipulated object as a single rigid body under differentiable physics, rather than optimizing each Gaussian independently.

### A.3.4    SCOPE, LIMITATIONS, AND PATH TO GENERAL POLICIES.

These results highlight the flexibility of our engine while clarifying its scope. Generalization currently depends on (i) accurate mesh reconstruction and mass identification and (ii) task setups whose contact conditions are well approximated by our rigid-body simulator. The present policies remain object-specific; however, conditioning on an estimated mass offers a plug-in signal that can be combined with architectures designed for category-level or multi-object training (e.g., Zhang et al. (2024c)) to obtain more general policies when suitable demonstrations are available. Grounding learning in human demonstrations and targeted parameter identification reduces reliance on hand-engineered rewards and large-scale robot-collected datasets, enabling data-efficient transfer across tasks.

Table 8: Cross-object generalization from a larger to a smaller electric screwdriver. The policy is trained on five human demonstrations and conditioned on object mass and reconstructed mesh.

| Training Object | Test Object | Success Rate |
|---|---|---|
| $10 \times 3 \times 3$ cm, 600 g | $7 \times 2 \times 2$ cm, 500 g | 90% → 70% |

### A.3.5    MASS-AWARE LEARNING VS. DOMAIN-RANDOMIZED RL

We compare D-REX—trained from human demonstrations and conditioned on accurately inferred object mass—against CrossDex Yuan et al. (2024), a reinforcement-learning baseline using domain randomization (DR). CrossDex randomizes mass in the range $0.5$–$1.5$ kg during training and reports an 89% success rate in simulation.

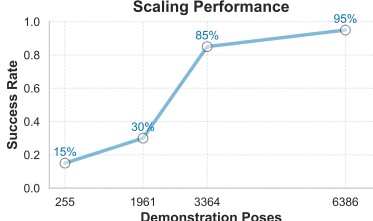
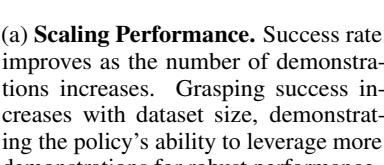
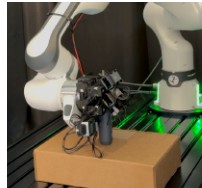
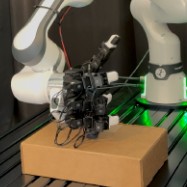
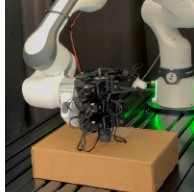

(a) **Scaling Performance.** Success rate improves as the number of demonstrations increases. Grasping success increases with dataset size, demonstrating the policy's ability to leverage more demonstrations for robust performance.

(b) **Visualization of Grasping Pose with respect to number of demonstrations.** As shown in the figures, the policy produces unstable grasps with 1 to 10 demonstrations, but generates a stable grasp when trained with 20 demonstrations.

Figure 9: Scaling performance of our force-aware grasping policy with increasing number of demonstrations.

To isolate the effect of mass, we evaluate on a family of *Symbol Y* objects that share identical geometry but differ in mass: $117\,g$, $206\,g$, and $324\,g$ (i.e., $0.117$, $0.206$, and $0.324\,kg$). Notably, all three test masses lie *below* the CrossDex training range. As summarized in Table 9, CrossDex performs well on the heaviest variant and moderately on the medium one, but struggles on the lightest object, illustrating DR's sensitivity when deployed on out-of-distribution (OOD) mass values, especially far from the training support.

In contrast, D-REX leverages low-cost human demonstrations to infer object mass and conditions the policy accordingly, enabling targeted adaptation without additional randomized training. The result is consistently high success across all three masses, despite the OOD shift relative to the DR baseline's training range.

Table 9: Real-world grasp success across mass variants of a single object geometry (*Symbol Y*); 10 trials per condition. CrossDex was trained with mass randomization in $[0.5, 1.5]\,kg$; all test masses are below this range. Higher is better.

| Method | 117 g (Light) | 206 g (Medium) | 324 g (Heavy) |
|---|---|---|---|
| CrossDex | 4/10 | 7/10 | 9/10 |
| Ours | 9/10 | 10/10 | 9/10 |

These results suggest two complementary points: (i) Domain Randomization can yield strong performance within or near its training support but degrades for larger OOD mass shifts (e.g., very light objects), and (ii) explicit, mass-aware conditioning provides a simple and data-efficient mechanism for robust transfer across mass variation without requiring broad randomization. While Domain Randomization and mass-aware learning are not mutually exclusive, our findings indicate that accurate parameter identification is a powerful lever for real-world generalization, particularly when deployment conditions fall outside the range covered by domain randomization.

### A.4 LEARNING PHYSICAL PARAMETERS BEYOND MASS AND HANDLING FRAGILE OBJECTS

We focus on estimating object mass because it admits a clear ground truth, is straightforward to validate experimentally, and exerts an immediate and observable influence on grasping performance (e.g., grasp failures due to underactuation). While it is in principle feasible to learn additional physical parameters—such as friction, stiffness, or damping—we prioritize mass owing to its measurability, stability across settings, and direct relevance to grasp dynamics. Moreover, mass variation can be applied systematically across diverse objects, which enables a controlled assessment of generalization across geometries and densities.

Prior work has explored learning richer sets of physical properties in simulation (e.g., gradSim Jatavallabhula et al. (2021b)) and dense object attributes from visual observations (e.g., Xu et al. (2019)). However, reliable real-world validation of such parameters remains substantially more challenging

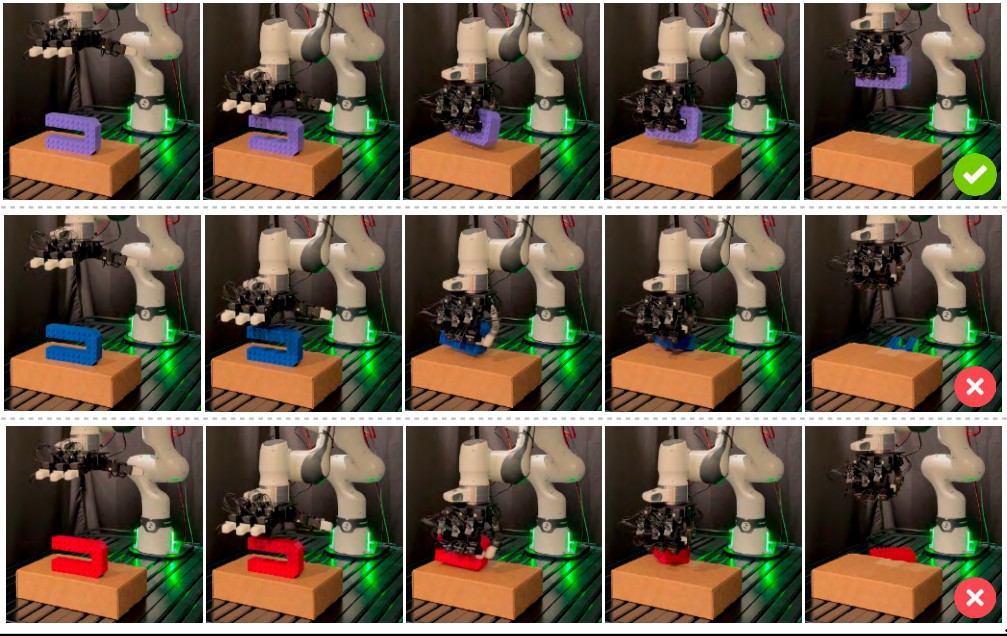

Figure 10: **Quantitative Results of Object Grasping trained on the heavy one.** Only the mass-matched policy achieves stable grasps, while mismatched ones fail due to excessive or insufficient force, causing bounce-off for lighter objects or slippage for heavier ones.

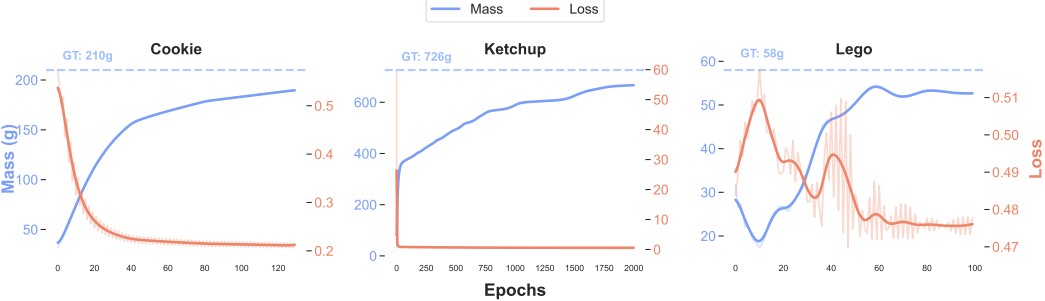

Figure 11: **Mass-Loss curves.** We present three examples of our system applied to mass identification. The blue curves represent the estimated masses, all of which converge reliably to the ground-truth values, demonstrating the accuracy of our approach.

due to contact dependence, spatial and temporal variability, and sensitivity to surface conditions. Consequently, extending parameter learning beyond mass is outside the scope of the present study.

In our current D-REX engine, rigid-body dynamics are assumed and object mass is the sole learnable physical parameter. This design choice serves two purposes: (i) it isolates the causal role of mass in dexterous grasping and (ii) it demonstrates that D-REX can recover this key quantity directly from data. The policy is conditioned on demonstrations for the same object, making it object-specific rather than fully general. For broader generalization, the mass-estimation module can be used as a plug-in: estimate an object's mass and then fine-tune (or condition) a task policy on the inferred value.

For objects that are too fragile to tolerate pushing, we do not attempt to learn additional contact parameters. Instead, we employ lower-force grasping strategies to reduce contact uncertainty—for example, rolling or reorientation while following a predefined orientation trajectory (via quaternion `slerp`) Xu et al. (2019)—thereby limiting impulsive interactions without requiring explicit estimation of frictional properties.

Finally, after the real-to-sim alignment step, execution proceeds without external human intervention in the physical scene. This assumption preserves consistency with the Lagrangian rigid-body dynamics underlying our simulator and cleanly attributes observed performance differences to the learned mass parameter rather than to uncontrolled external corrections.

## A.5 ON THE GENERALIZATION OF D-REX

As noted in our limitations, the current policy is object-specific: training and evaluation assume that the object's mass is consistently identified and transferred between simulation and the real system. Nevertheless, we examine the extent to which D-REX exhibits cross-object and cross-task generalization under this assumption.

**Within-category, cross-object transfer.** To assess transfer within a category, we collected twenty human demonstration episodes for a larger electric screwdriver and trained a policy conditioned on its reconstructed mesh and mass. At test time, *without any fine-tuning*, we replaced the mesh and mass with those of a smaller screwdriver from the same category and executed the policy for 10 trials. As summarized in Table 8, the policy maintained stable performance with only a minor decrease in success rate, indicating that D-REX generalizes across moderate variations in geometry and mass within a category. Qualitative rollouts are provided on the anonymous project website.

**Beyond grasping: articulated and fine-grained tasks.** To probe broader applicability, we evaluated D-REX on more complex grasping scenarios, including articulated-object interactions (e.g., operating a stapler) and fine-grained tasks (e.g., manipulating a computer mouse). For each task, we used 5–10 human demonstrations processed through the same real-to-sim pipeline, but for the articulated object digital asset creation, we manually perform the segmentation and re-assembly, with mass identification integrated into training. We find that, provided the reconstructed simulation captures the salient articulated structure and task-relevant geometry, the policy transfers reliably to these settings, suggesting that D-REX is not limited to grasping but can support a wider class of object interactions.

## A.6 PERFORMANCE DEGRADATION OF THE BASELINE

The primary cause of the baseline's degraded performance is the absence of explicit force (or impedance) control combined with a narrow training mass distribution. Both *Human2Sim2Robot* and *DexGraspNet 2.0* Zhang et al. (2024c) were trained entirely in simulation with object masses concentrated around $\approx 0.1\,\mathrm{kg}$. When deployed on substantially heavier items (e.g., *Spam*, *Ketchup*, *Nutella*), which lie outside this training distribution, the controller applies essentially fixed or weakly adaptive grasp forces that are insufficient to prevent slip—i.e., grasp failures due to underactuation.

For a gravity-resisting, frictional grasp, the required normal force per contact grows with the object weight and inversely with the effective friction coefficient. In a simplified parallel-jaw setting with two symmetric contacts,

$$F_n \gtrsim \frac{mg}{2\mu}\,\gamma,$$

where $m$ is the object mass, $g$ is gravitational acceleration, $\mu$ is the effective (task-dependent) friction coefficient, and $\gamma \geq 1$ absorbs wrench distribution, contact geometry, and safety margins. If $m$ is outside the range encountered during training—or is underestimated at deployment—a fixed position-control policy lacks the ability to scale $F_n$ accordingly, violating the inequality and inducing slip.

Low-friction surfaces (e.g., plastic wrap or smooth metal) further increase the force requirement by reducing $\mu$, exacerbating failures when the applied force is already marginal. Conversely, lighter objects are more tolerant to small positioning or force errors and may remain secured despite suboptimal control. We also observe exceptions where heavier objects succeed due to fortuitous geometry: for instance, spray-bottle nozzle heads can incidentally create partial form-closure (or caging) between fingers, partially compensating for insufficient frictional support.

Our method augments the policy with mass-aware force modulation: we estimate object mass from robotic action and videos and adjust the grasp force (or impedance setpoints) as a function of the inferred mass at test time. This targeted adaptation restores adequate contact forces on heavier

or otherwise challenging objects, reducing slip and improving success rates. More broadly, these findings underscore the necessity of mass-conditioned control for robust, generalizable dexterous grasping across diverse real-world objects and surface conditions.

### A.7 REASON TO BUILD UP ACCURATE DIGITAL TWIN

Building accurate digital twins and applying domain randomization are two complementary strategies for bridging the sim and real gap, each offering distinct advantages depending on the task and deployment context. Accurate digital twins aim to faithfully reproduce real-world physical and visual fidelity, etc. enabling: 1) Precise policy evaluation and benchmarking under realistic dynamics, 2) System identification, particularly for contact-rich tasks or sensitive physical parameters such as mass and friction, 3) Gradient-based optimization of physical properties or control strategies, which requires differentiable and realistic simulation feedback.

Our approach extends beyond visual or geometric digital twins by incorporating differentiable system identification to capture underlying physics—a long-standing challenge in robotics and graphics. This enables more accurate and efficient parameter adaptation, improving both realism and policy transfer. We view digital twins and domain randomization as complementary tools, with high-fidelity modeling serving to support informed adaptation in contact-rich or dynamic scenarios where randomization alone may overlook critical constraints.

### A.8 RELATIONSHIP BETWEEN MASS IDENTIFICATION AND FORCE-BASED POLICY LEARNING

We deliberately decouple *mass identification* from *policy learning* to isolate the causal role of mass in sim-to-real transfer and to enable clean evaluation. Concretely, from a small set of human demonstrations and robot grasping $\mathcal{D}$ we estimate a scalar mass

$$\hat{m} \;=\; \arg\min_{m} \; \mathcal{L}_{\mathrm{id}}(m; \mathcal{D}),$$

and then train a control policy that is explicitly conditioned on this estimate,

$$u \;=\; \pi_{\theta}(x, \hat{m}),$$

where $x$ denotes the robot/object state and $u$ the control command. This two-stage design avoids confounding between parameter estimation and control optimization, making it possible to attribute downstream performance changes specifically to the accuracy of $\hat{m}$.

Conditioning the policy on $\hat{m}$ enables explicit modulation of force or impedance setpoints and of feedforward gravity terms (e.g., $u \supset g(q; \hat{m})$). In practice, we scale grasp-force targets and compliance parameters as functions of $\hat{m}$, which restores adequate contact forces on heavier objects while avoiding unnecessarily high forces on lighter ones. The result is improved robustness across a broad mass range without requiring extensive re-training.

### A.9 SYSTEM SUBMODULES AND LIMITATIONS

Prior differentiable real-to-sim approaches (e.g., Chen et al. (2025b)) typically rely on rich robot proprioception (e.g., motor torque sensing) and tight hardware calibration to perform system identification. Such requirements limit deployability outside well-instrumented labs and differ substantially from our setting. By contrast, we pursue *vision-driven* identification that uses only externally observed signals, which we find more accessible and scalable in practice. Accordingly, we evaluate against *ground-truth physical measurements* (e.g., mass) rather than sim-only metrics, providing a direct assessment of real-world fidelity. To our knowledge, few real-to-sim engines offer end-to-end differentiability that remains practical at deployment time; those that do often require assumptions that are difficult to satisfy in unstructured environments.

Our pipeline leverages *FoundationPose* Wen et al. (2024) as a robust 6-DoF pose estimator. These poses serve as the primary observation signal for identification and control, replacing the need for onboard torque sensing. We combine these estimates with a differentiable physics engine (MJX) operating on real2sim-generated MJCF assets, which supply geometry, inertial properties, and contact models.

Gradsim Jatavallabhula et al. (2021b) is designed for System Identification using rendered image observations from simulation, combining state-based and photometric losses. Our setting differs in two fundamental ways:

**Real-world, partial observations.** We operate directly on real videos and 6-DoF object poses estimated by FoundationPose Wen et al. (2024). Rather than assuming access to full simulator state and gradients as in Jatavallabhula et al. (2021b), we estimate physical parameters (e.g., mass) from *partial*, noisy observations by minimizing state-space trajectory error over time in a differentiable simulator.

**Photometric supervision is impractical in our setup.** Applying Jatavallabhula et al. (2021b) would require carefully controlled lighting, calibrated cameras, and often even 3D-printed objects with known properties to obtain reliable photometric losses. We explored using 4D Gaussian Splatting to synthesize renders for photometric alignment, but optimization was unstable and inaccurate in our scenes, reinforcing the limitations of purely image-based losses for physical deployment.

**Physics-constrained identification.** In our engine, the differentiable simulator acts as a numerical solver obeying physical laws. Given known robot inputs (e.g., commanded joint trajectories) and accurate initial/boundary conditions from Real2Sim, we pose mass estimation as a constrained optimization problem: find the parameter values that best reproduce observed FoundationPose trajectories.

For these reasons, we do *not* treat Jatavallabhula et al. (2021b) as a competing baseline in our evaluation. Instead, we reuse its differentiable rendering mechanism internally while MJX provides the physical kinematic and dynamic of robotic hand.

**Modularity, robustness, and extensibility.** Our system is intentionally modular: (i) pose estimation (FoundationPose Wen et al. (2024)); (ii) asset generation and scene reconstruction (Real2Sim) Lou et al. (2024); Ye et al. (2024b;c); (iii) differentiable physics (MJX,Gradsim) Jatavallabhula et al. (2021b); Todorov et al. (2012); and (iv) policy learning. Similar multi-component designs are common in robotics and vision system work Pfaff et al. (2025); Andrychowicz et al. (2020) because they enable targeted improvements, swapping of submodules as better tools emerge, and reusability of well-validated components. We do not treat these modules as black boxes; rather, we select them based on empirical reliability and integrate them with sanity checks and data filtering.

**Data strategy and practicality.** We rely on human demonstration videos as the primary supervision signal for policy learning. Such videos are inexpensive and widely accessible (e.g., public internet platforms and existing datasets), dramatically reducing the collection burden compared to robot-executed demonstrations or reinforcement learning, which often require hand-engineered rewards and long training cycles. Occasional failures of individual submodules (e.g., transient pose estimation errors) typically result only in filtering out a small fraction of low-quality demonstrations; overall effectiveness is maintained through scale. The combination of scalable supervision with differentiable real-to-sim identification yields a practical and extensible pathway toward robust sim-to-real transfer.

**Limitations.** The D-REX framework currently only supports rigid-body dynamics and relies solely on mass as the primary learnable parameter. Once the real-to-sim stage concludes, our simulation engine requires the absence of human interaction with the real-world operation scene to maintain consistency under the assumed Lagrangian dynamics framework.

## A.10   LLM USAGE

We employed large language models solely for grammatical refinement and stylistic polishing of the manuscript. No part of the conceptualization, experimental design, implementation, or analysis relied on these models.

## A.11 LIST OF NOTATIONS

| Symbol | Description |
| --- | --- |
| $I_s$ | Scene-centric RGB video sequences |
| $I_o$ | Object-centric RGB video sequences |
| $\{I_t\}_{t=1}^T$ | Human demonstration RGB video sequences |
| $\{s_t^{\text{real}}\}_{t=1}^T$ | Real-world object trajectories |
| $\{s_t^{\text{sim}}\}_{t=1}^T$ | Simulated object trajectories |
| $m$ | Optimized object mass |
| $\pi$ | Force-aware grasping policy |
| $S$ | Simulation environment representation (MJCF) |
| $K$ | Collision mesh for object geometry |
| $\theta$ | Physical simulation parameters |
| $P$ | Gaussian splatting particles for visual appearance |
| $s_t$ | Object's state at timestep $t$ (position and orientation) |
| $u_t$ | Object's velocity at timestep $t$ |
| $v_t$ | Linear velocity component at timestep $t$ |
| $\omega_t$ | Angular velocity component at timestep $t$ |
| $M$ | Mass-inertia matrix |
| $f$ | External and contact forces |
| $f_n$ | Contact force vector |
| $k_e, k_d$ | Contact stiffness and damping parameters |
| $G(\cdot)$ | Discrete-time update function |
| $\Delta t$ | Simulation timestep |
| $L_{\text{traj}}$ | Trajectory loss function |
| $h_t$ | Human hand pose at timestep $t$ |
| $o_t$ | Object pose at timestep $t$ |
| $A_t$ | Robot action at timestep $t$ |
| $\pi_\phi$ | Learned grasping policy (parameterized by $\phi$) |
| $\hat{A}$ | Predicted robot joint positions |
| $\hat{r}$ | Predicted contact constraint |
| $\hat{f}$ | Predicted grasping force constraint |
| $n_{\text{active}}$ | Number of active contacts between robot and object |
| $\rho$ | Object density parameter |

