# OpenReview forum: "D-REX: Differentiable Real-to-Sim-to-Real Engine for Learning Dexterous Grasping"
_ICLR.cc/2026/Conference — ICLR 2026 Poster_

### Official Review · Reviewer_KAq7 · 2025-10-25

**Soundness:** 3
**Presentation:** 2
**Contribution:** 3
**Rating:** 6
**Confidence:** 4

**Summary:**

The paper proposes D-REX, a real-to-sim-to-real method to improve the performance of grasping policies. The method first leverages off-the-shelf tools to convert human videos to robot kinematic trajectories. After that, the method identifies object masses based on differentiable physics engines. The identified object masses are further utilized to develop a force-based control policy for grasping. Extensive experiments prove the effectiveness of the method. Overall, the paper makes a step forward in developing physics-aware policies utilizing real-to-sim-to-real methods and demonstrates the effectiveness of incorporating physical properties into robot policies. Main limitations lie in the inherent restrictions of the mass identification method and the restricted upper limits of only identifying and utilizing masses in the robot policy learning.

**Strengths:**

- Good motivations. The problem is well motivated. Real-to-sim system identification is an important way to bridge the sim-to-real gap. Beyond properties reflected from the visual appearances, such as object meshes, the paper makes a step forward and proposes to identify physical properties, i.e., masses, from a dynamic interaction sequence. After identifying masses, a force-adaptive method is developed to improve the grasping policy.
- Reasonable methodology. Estimating masses from hand-object interaction sequences and the force-based policy are reasonable approaches.
- Solid experiments. The authors carefully design experiments to validate the effectiveness of the object mass identification method design and the superiority of the force-based grasping policy.

**Weaknesses:**

- The paper utilizes the foundation pose to estimate the object pose sequences from real videos. However, the foundation pose cannot deal with axis-symmetric objects and tiny objects. Moreover, the estimated object poses always suffer from noise. Therefore, the quality of the mass identification step would be restricted by the quality of the identified masses. The applicability of the method would also be restricted to objects that the foundation pose can handle. Besides, utilizing foundation models to generate scene configurations, such as the robot mjcf, may also introduce errors.
- When identifying masses from videos, the sim-to-real gap in other properties, such as frictions, is neglected, which would further make the estimation prone to errors.

**Questions:**

- How do you train the grasping position policy?
- Could the method be extended to small objects, including both the mass identification and the grasping process?

---

> ### Author Response · Authors · 2025-11-22
>
> > **Q1**: Error raised by FoundationPose and VLM based MJCF generation
>
> **A1**: We appreciate the reviewer’s concern regarding the limitations of current 6-DoF pose estimators such as FoundationPose and small objects. In our system, mass identification is explicitly trajectory-based rather than relying on single-frame pose estimates. Masses are optimized over complete pose–force trajectories, which makes the estimation robust to moderate pose noise and occasional outliers.
> Pose noise produced by FoundationPose is relatively small and the optimization process is able to smooth out frame-wise errors. This design follows common practice in pose-based manipulation pipelines[1]. Importantly, our mass identification algorithm is agnostic to the specific pose estimator: if a more accurate or specialized estimator for symmetric or small objects becomes available, it can be directly plugged into our pipeline and would only improve performance.
> For the role of VLM- and foundation-model–based Real2Sim in our framework. These models are used to generate an initial scene configuration, including the robot MJCF and object meshes, in order to reduce manual modeling effort. During mass identification, this scene description is kept fixed and mass is the only physical parameter that is varied across experiments. As a result, small modeling inaccuracies in the MJCF, such as slight deviations in link dimensions or inertia tensors, act as a constant bias and do not affect the relative effectiveness of our mass identification procedure or the conclusions drawn from our experiments. Moreover, when higher-fidelity models are needed, users can optionally refine MJCFs by hand, as we note in the paper.
> For the impact of imperfect mass identification on downstream grasping, we provide an empirical evaluation in Table 5. These results show that the grasp success rate degrades only mildly under realistic mass estimation errors, indicating that the downstream grasping policy is tolerant to the residual inaccuracies in mass. In other words, perfect mass identification is not required to achieve the level of grasping performance we report. While foundation pose estimators and automatic scene generation are not error-free, our method is designed to be robust to the resulting moderate noise, and our experimental protocol controls other confounding factors so that the identified masses remain reliable for downstream control.
>
> > **Q2**: Physical parameter identification other than mass for sim-to-real gap
>
> **A2**: Regarding the sim-to-real gap in other physical properties such as friction, accurately identifying these parameters is very challenging with current hardware and experimental setups. It typically requires specialized sensors and carefully controlled experiments, which lies outside the scope of this work. In this paper, we therefore focus on mass as the primary unknown parameter and treat other quantities, such as friction coefficients, as fixed but reasonably tuned in the simulator. We view explicit friction identification as an important but orthogonal direction for future work, rather than a prerequisite for demonstrating the benefits of Real2Sim mass identification.
> Extending our approach to jointly identify mass, inertia, friction, and joint compliance is a promising direction. D-Rex is compatible with such multi-parameter identification, provided that the observations can be modeled accurately and that experiments are carefully designed.

---

> ### Author Response · Authors · 2025-11-22
>
> > **Q3**:   Training details of the grasping position policy
>
> **A3**:  For the grasp position policy, we use a two-stage training procedure that combines human demonstrations with simulation-based refinement (details in Appendix A.2.4). Our grasping policy, denoted **GraspMLP**, is first trained from human data and then refined in simulation.
> We start by collecting human grasp demonstrations and reconstructing human hand and object poses using HaMeR[2] and MCCHO[3]. These human hand trajectories are retargeted to the target robotic hand, giving joint-space reference actions. For each object, we use the reconstructed mesh from Real2Sim together with its identified mass. We sample mesh vertices, apply positional encodings to them, and concatenate these features with the object mass to form the input to GraspMLP.
> GraspMLP has a shared MLP backbone that processes the per-vertex 3D coordinates and mass-conditioned features. On top of this backbone, we use three heads: one for joint actions (16D), one for a contact-based reward/feasibility score (2D), and one for a normalized grasp force (1D). The full architecture is listed in Table 4.
>
> Training has two phases:
>
> Phase 1 (supervised pre-training): We train **GraspMLP** on human demonstrations to regress joint actions and predict contact/force labels. We use MSE loss for joint actions and force, and BCE loss for contact prediction (Algorithm 2, lines 3–13). All demonstrated grasps are treated as successful in this stage, and contact/force labels are set accordingly.
>
>  Phase 2 (simulation refinement): We roll out the policy in MuJoCo using the Real2Sim-generated MJCFs and the identified object masses (Algorithm 1, lines 6–7; Algorithm 2, lines 15–26, in Appendix). From these rollouts we compute a contact-based reward that measures grasp success and stability, and a scaled force target based on object mass, gravity, and number of contact points. We then update GraspMLP with a weighted combination of BCE (for rewards) and MSE (for force targets), effectively doing RL-style fine-tuning within a supervised-learning setup.
>
> At deployment time, the policy takes the observed object point cloud (same format as the training vertices) and the identified mass as input. On real hardware, we implement a force-based controller by mapping the predicted grasp force to actuator currents. For Allegro and LEAP hand, which use direct-drive brushless motors, this mapping is approximately linear, so current control is a good proxy for fingertip force.
>
> > **Q4**:  Generalization of D-Rex framework
>
> **A4**:  Our method is not limited to the object sizes used in our experiments. Both the mass identification and grasping modules operate on pose trajectories and reconstructed geometry, so in principle they can be applied to smaller objects as well.
> With current hardware and perception foundation models, handling very small objects is mainly limited by two factors: the perception pipeline must still provide accurate 6-DoF poses and geometry, and the robot hardware (including end-effector design and finger size) must be capable of reliably grasping and manipulating such objects. In our current setup, these perception and hardware constraints are the main bottlenecks for tiny objects, rather than a limitation of the Real2Sim or mass identification formulation itself. These challenges are shared by most pose-based manipulation methods. In this paper, we therefore focus on object sizes that are standard in existing 6-DoF pose estimation benchmarks and robotic grasping setups, where both perception and hardware are reliable.
>
>
> We are very grateful for the reviewers’ detailed comments and suggestions, which we believe will help us further improve and strengthen this work.
>
> [1]Hermes: Human-to-robot embodied learning from multi-source motion data for mobile dexterous manipulation, arxiv 2025
>
> [2]Hamer:Reconstructing Hands in 3D with Transformers, CVPR 2024
>
> [3]MCC-Ho: Reconstructing Hand-Held Objects in 3D
> from Images and Videos, arxiv 2024

---

> ### Author Response · Authors · 2025-11-27
> **Thank you for your reviews**
>
> Dear reviewer KAq7,
>
> Thank you again for providing the very constructive review!
>
> Since the end of the rebuttal period is approaching (Dec 03, 9:00 PM UTC), we would like to kindly follow up to check if the provided responses have sufficiently addressed your questions and concerns. If so, we kindly hope that you might be willing to raise the level of your recommendation. Thanks again! :)
>
> Best, Authors of paper 333

---

### Official Review · Reviewer_M426 · 2025-10-27

**Soundness:** 3
**Presentation:** 3
**Contribution:** 3
**Rating:** 6
**Confidence:** 1

**Summary:**

The paper presents a differentiable real-to-sim-to-real engine for object mass identification and grasping policy learning. Specifically, the Gaussian Splat representation is leveraged to facilitate estimation object mass through visual observations and robot control signals during interaction. Besides, a learning-based method is also proposed to train force-aware grasping policies from limited human demonstration videos. Comprehensive experiments have been conducted to validate the performances on mass identification and object grasping.

**Strengths:**

1.	The topic of end-to-end object mass identification is valuable and the solution using differentiable simulation is novel.
2.	Based on the mass estimation, the proposed system achieves satisfactory performances on the object grasping, reducing the gap between simulator and real-world environment.
3.	The proposed approach outperforms strong baselines in object grasping, especially for the challenging object grasping.
4.	The paper is well-written and the ablation studies are comprehensive.

**Weaknesses:**

1.	There should more details in the section of parameter identification from robot-object interactions. For instance, the rationale behind the trajectory discrepancy minimization for the object mass identification should be included. What’s the effects of the semi-implicit Euler modeling. Is there any ablation study for this learning objective?

**Questions:**

Please see the weaknesses.

---

> ### Author Response · Authors · 2025-11-22
>
> > **Q1**:Details of parameter identification
>
> **A1**: We thank the reviewer for pointing out the need for more details on the mass identification objective.
>
> We  expand the parameter-identification section to provide a more formal ODE-based description of our objective, in terms of boundary,initial conditions, sampling, and convergence.
>
> The object dynamics in our setup are modeled by the Newton--Euler equations
> \begin{equation}
>     m \ddot{x}(t) = f_{\text{contact}}(x(t), u(t)) + f_{\text{grav}},
> \end{equation}
> where $x(t)$ denotes the object pose (or position of its center of mass), $u(t)$ encodes the robot hand actions, and $m$ is the unknown object mass.
>
> The initial conditions and geometric constraints are determined by:
>
> (i) the object geometry and pose reconstructed via Gaussian splatting (initial pose/shape),
>
> (ii) the robotic hand configuration and known actuation commands (control input $u(t)$),
>
> (iii) the ground plane and environment, also provided by the reconstruction.
>
> The dynamics residual is evaluated at discrete time steps  $\{t_k\}$ along the observed trajectory, with contact forces computed from the mesh vertices of the Gaussian-splatting reconstruction.
>
> The ground-truth trajectories $\{\hat{x}_k\}$ at discrete times $\{t_k\}$ are provided by FoundationPose. Our differentiable simulator enforces the governing Newton--Euler equations at these sampled points, so that convergence of the learned dynamics and mass parameter is driven by minimizing the residual of a well-posed ODE system.
>
>
> In the specific ''push-down'' interaction considered for mass identification, let $z(t)$ denote the object position along this axis, and $u(t)$ the known contact force applied by the robot:
> \begin{equation}
>     m \ddot{z}(t) = u(t) - m g.
> \end{equation}
> Integrating twice in time, we obtain
> \begin{equation}
>     z(t; m) = z_0 + v_0 t + \frac{1}{m}\int_0^t \int_0^{\tau} u(s)\,ds\,d\tau - \frac{1}{2} g t^2,
> \end{equation}
> which can be written as
> \begin{equation}
>     z(t; m) = \alpha(t) + \frac{1}{m}\\cdot\beta(t),
> \end{equation}
> where $\alpha(t)$ and $\beta(t)$ depend only on known quantities (initial conditions and measured robot forces). Thus, under this simplified contact model, the trajectory is affine in the inverse mass $1/m$. Discretizing time gives
> \begin{equation}
>     z_k(m) = \alpha_k + \frac{1}{m}\\cdot\beta_k,\quad k=1,\dots,T.
> \end{equation}
> The trajectory loss we use,
> $$
> L_{traj}(m) = \\sum_{k=1}^T ( z_k(m) - \\hat{z}_k )^2
> $$
>
> then becomes a standard least-squares problem in the parameter $\theta = 1/m$. In this parametrization, $\mathcal{L}_{\text{traj}}(\theta)$ is a quadratic function of $\theta$ and therefore convex. In practice, our learning procedure operates through vector--Jacobian products in the differentiable simulator, but the underlying structure is equivalent to solving a well-conditioned least-squares identification problem for the mass.
>
> We add this analysis to the appendix to explicitly justify why minimizing trajectory discrepancy is a principled way to identify mass in our setting, and why it leads to a numerically stable optimization landscape.

---

> ### Author Response · Authors · 2025-11-22
>
> > **Q2**: Ablation study on semi-implicit Euler against explicit Euler
>
> **A2**: We agree that the effect of the semi-implicit Euler scheme should be clarified.
>
> For the discrete-time dynamics, we use a semi-implicit Euler scheme. For a time step $\\Delta t$, the updates are
>
> $$
> v_{k+1} = v_k + \\Delta t \\cdot m^{-1} \\cdot f(x_k, u_k),
> $$
>
> $$
> x_{k+1} = x_k + \\Delta t \\cdot v_{k+1},
> $$
>
> where $f(x_k, u_k)$ includes gravity and contact forces. In contrast, the explicit Euler scheme would use
>
> $$
> v_{k+1} = v_k + \\Delta t \\cdot m^{-1} \\cdot f(x_k, u_k),
> $$
>
> $$
> x_{k+1} = x_k + \\Delta t \\cdot v_k.
> $$
>
> following the modeling of PBD[1].
>
> In contact-rich dynamics, especially for small masses or stiff contact forces, explicit Euler has a much smaller stability region and tends to produce large, oscillatory updates in $x_k$. This is particularly problematic in our setting because we initialize the mass at a relatively small value (a conservative lower bound), which amplifies the effective acceleration $m^{-1} f$ and can lead to very large position updates and exploding gradients when using explicit Euler. Semi-implicit Euler, on the other hand, updates the velocity first and then uses the updated velocity to advance the position.
>
> Empirically, the explicit Euler method is computationally faster but less accurate than the semi-implicit Euler method.
> | Object | Ground truth (g) | Semi-implicit (g) | Explicit (g) | Semi-implicit (s) | Explicit (s) |
> |--------|------------------|-------------------|-------------|--------------------|--------------|
> | Lego   | 59               | **51**                | 34       | 1.38               | 1.21         |
> | Ketchup| 726              | 667              | **685**        | 1.43               | 1.19         |
> | Cookie | 210              | **200**               | 189         | 1.41               | 1.18         |
> | Domino | 106              | **117**               | 135         | 1.36               | 1.17         |
> | U      | 125              | **110**               | 98          | 1.39               | 1.20         |
> | A      | 134              | **145**               | 120         | 1.42               | 1.22         |
>
> We sincerely thank the reviewer for their thoughtful comments and constructive feedback, which have helped us better clarify our method and strengthen the overall paper.
>
> [1]Position based dynamics, Journal of Visual Communication and Image Representation 2007

---

> ### Author Response · Authors · 2025-11-27
> **Thank you for your reviews**
>
> Dear reviewer M426,
>
> Thank you again for providing the very constructive review!
>
> Since the end of the rebuttal period is approaching (Dec 03, 9:00 PM UTC), we would like to kindly follow up to check if the provided responses have sufficiently addressed your questions and concerns. If so, we kindly hope that you might be willing to raise the level of your recommendation. Thanks again! :)
>
> Best, Authors of paper 333

---

### Official Review · Reviewer_6NV8 · 2025-11-01

**Soundness:** 3
**Presentation:** 3
**Contribution:** 4
**Rating:** 6
**Confidence:** 5

**Summary:**

The paper presents D-REX, a differentiable real-to-sim-to-real framework for dexterous grasping. It combines 4D Gaussian Splatting (4DGS) for scene reconstruction and differentiable physics for identifying physical parameters (notably object mass) from real-world robot trajectories. The identified mass is then used to train a force-aware grasping policy, which achieves higher grasp success rates in both simulation and limited real-world settings.


The paper explores an interesting and relevant direction by combining differentiable physics and 4DGS for Real2Sim2Real learning. While the current experiments do not fully validate the motivation of building a high-fidelity differentiable Real2Sim pipeline, the proposed idea is novel and potentially impactful. Overall, the paper presents a promising step toward differentiable Real2Sim learning and merits a weak accept pending more thorough validation.

**Strengths:**

- The paper is clearly written and presents a coherent overall system.
- The proposed framework is conceptually appealing, and incorporating mass identification for robot policy learning is a novel and promising idea.
- The experimental results demonstrate accurate mass estimation and consistent grasping improvement with mass-aware policies.

**Weaknesses:**

- **The evaluation of Real2Sim quality is lacking.**
  The paper presents no analysis or evaluation of either appearance or geometry (mesh) of the generated digital scenes—offering neither quantitative metrics nor qualitative discussion. As a result, the claimed Real2Sim objective remains unsupported and unvalidated, which weakens the overall completeness and credibility of the contribution.

- **The validation of the “force-aware policy” is insufficient.**
  To validate the effectiveness of the force-based control, the authors report grasping success rates in Table 3 and Figure 5 undering different settings. A visual or quantitative comparison of force values across different settings during robot execution would better substantiate the effectiveness of the proposed force-aware policy learning.

- **The efficiency and scalability should be discussed.**
  The authors present mass-loss curves in Figure 11 but seem to have omitted the actual optimization or training time, which is also an important factor for evaluating system efficiency. In addition, the offline 4DGS reconstruction step can be computationally expensive; however, no quantitative analysis (e.g., runtime or memory usage) is provided. Therefore, the practicality of the proposed Real2Sim pipeline for large-scale or online applications remains somewhat unclear.

- **The generalization ability of D-REX is limited.**
  The proposed pipeline requires real-world object trajectories (obtained via FoundationPose) and matched real/sim robot interactions to optimize object mass. Consequently, the generalization ability of the proposed method to novel scenes or objects, where such real trajectory data and robot interactions are unavailable, appears limited.

**Questions:**

1. The paper introduces two separate sets of Gaussians to represent the visual appearance and geometry of the scene. How might potential optimization misalignment between these two representations affect the accuracy of mass identification and the subsequent policy learning?
2. The Appendix mentions that the optimization of 4D Gaussian Splatting for photometric alignment is unstable and inaccurate. Could you provide detailed explanations or illustrative failure examples to clarify this issue?

## Suggestions for Improvements

1. Provide more qualitative and quantitative results/comparisons after Real2Sim stage, evaluating both the 4DGS scenes and the generated mesh.
2. Provide more visual or quantitative results/comparisons of force values during robot excution to better substantiate the force-aware policy.
3. Provide a systematic runtime analysis for different modules of D-REX to better assess the overall computational efficiency of the pipeline.
4. Minor formatting issues: Line 97: “Empirically, We” — the word “We” should be lowercase; Line 1737: “Physics-constrained identification” — a line break is recommended for proper formatting.

---

> ### Author Response · Authors · 2025-11-22
>
> We thank the reviewer for the careful reading and constructive suggestions. Below we provide a more detailed and structured response to each weakness, question, and suggestion in a continuous narrative.
>
>
> > **Q1**: Evaluation of Real2Sim quality (Weakness 1, Suggestion 1):
>
> **A1**: We thank the reviewer for raising this concern. To illustrate the performance of our reconstruction results, we show:
>
> 1) the reconstructed 4D scene rendered in our anonymous website  (section “Mass Identification via Object Pushing”).  The rendered video closely resembles the original video data;
>
> 2) the reconstructed geometry, visualized as an interactive 3DGS demo on our anonymous website (section “Interactive 3D Workspace”), where you can examine the reconstructed geometry by viewing the reconstructed objects at arbitrary viewpoints.
>
> In the revised version, we add a Real2Sim baseline comparison between our D-REX pipeline and vanilla 4D Gaussian Splatting (4DGS)[7] on the push-ketchup sequence. We evaluate along the robot’s trajectory using standard photometric metrics (PSNR, SSIM, LPIPS) and report inference runtimes:
> | Method | Optimization runtime | PSNR ↑ | SSIM ↑ | LPIPS ↓ |
> | ------ | ----------------- | ------ | ------ | ------- |
> | 4DGS   | 30 min            | 17.14  | 0.632  | 0.411   |
> | D-REX(3dgs + Optimized trajectory)  | **20 min**         |  **23.54**  |  **0.783**  |  **0.237**   |
>
> D-REX improves all three metrics, and we include these quantitative results to substantiate the Real2Sim quality. For implementation, we follow the standard D-NeRF bouncing ball dataset setting. Our method surpasses vanilla 4DGS because the supervision is applied directly on the 6-DoF object pose, and the ketchup object is modeled as a single rigid body, whereas 4DGS optimizes each Gaussian point independently.
> For the mesh reconstruction quality, we do not have the ground truth mesh of the object we reconstructed, so it is hard to compare it quantitatively. For the qualitative result, we will open source our dataset, including reconstructed meshes upon acceptance.
>
> > **Q2** : Answering the 4DGS difference with D-Rex 3dgs+optimized mass guided trajectory format (Question 2, Suggestion 1)：
>
> **A2**:  We thank the reviewer for pointing this out. In the appendix, we state that directly optimizing a 4DGS in our setting is “unstable” and “inaccurate”. By “unstable,” we mean that our target scenarios involve robotic arms and manipulated objects that lack texture in both appearance and geometry. This makes it hard for 4DGS to find valid visual correspondences and can cause the number of Gaussians to be out of control. By “inaccurate”: this also creates lots of floating Gaussians in the freespace, which presents inconsistent results in the rendered images.
>
> Specifically, 4DGS trained with only pixel losses produces (a) ghosting around the moving object, (b) temporally inconsistent shapes when the object is partially occluded. In contrast, our method uses 6-DoF poses from FoundationPose as initialization and optimizes object mass and trajectories in 3D space under rigid-body dynamics. This stabilizes optimization, yields smoother trajectories that are more consistent with physics, and is crucial for accurate mass identification.
>
> On our non-standard 4D datasets with fast object motion and static–dynamic hybrid scenes, such optimization often yields good RGB rendering on training views but inaccurate geometry along the depth (z-buffer) axis due to projection ambiguities. This causes notable rendering errors at novel viewpoints. Our differentiable physics module operates directly in 3D space using 6-DoF object poses estimated by FoundationPose, which yields more stable and physically meaningful trajectories than approaches that rely solely on pixel-space losses. In the final version, we will explicitly discuss and visualize the limitations of purely photometric optimization to better motivate our Real2Sim design.

---

> ### Author Response · Authors · 2025-11-22
>
> > **Q3**:  Clarification of the two Gaussian sets ( regarding Question 1)：
>
> **A3**: Our pipeline uses two separate Gaussian ensembles with distinct roles. The first ensemble, the geometry Gaussians (2DGS), is post-processed to obtain watertight meshes with fewer floaters. This ensemble is used exclusively for mass identification and physics-based optimization; the mass optimization depends only on this watertight geometry and the object’s 6-DoF trajectory. The second ensemble, the appearance Gaussians (3DGS), is optimized for high-quality rendering and visual realism. This appearance ensemble is not used in the differentiable physics optimization and does not influence mass estimation or policy learning. This design follows Drawer [1], where jointly optimizing geometry and rendering quality in a single Gaussian representation is often impractical in complex scenes.
>
> Because both ensembles are trained from the same multi-view data (with shared camera poses and initial point clouds), their difference reduces to a rigid scale and coordinate transformation. Following Robo-GS [4] and SplatSim [5], we align the ensembles via a simple scale-plus-rigid-transform step. Once aligned, the physics-optimized trajectories in the 2DGS/mesh frame are passed to the 3DGS ensemble for 4D rendering, similarly to PhysGaussian [6]. Any residual noise in the 3DGS ensemble affects only visual quality; mass identification and policy learning are computed solely on the watertight mesh and 6-DoF trajectories.
>
>
>
> > **Q4**:Validation of the force-aware policy (Weakness 2, Suggestion 2) ：
>
> **A4**: We appreciate the suggestion to directly visualize and quantify force signals. Our policy outputs a normalized force factor in $[0,1]$, which is then scaled by a fixed maximum current (corresponding to approximately 15 active motors). In the revision, we will add a table that summarizes, for each object, the inferred mass and the average per-motor current during successful grasps:
>
> | Object    | Inferred mass (g) | Avg. current per motor (mA) |
> | --------- | ----------------- | --------------------------- |
> | Domino    | 117               | 55                          |
> | Ketchup   | 667               | 102                         |
> | A         | 145               | 63                          |
> | U1        | 95                | 45                          |
> | U2        | 110               | 53                          |
> | U3        | 207               | 60                          |
> | Lego      | 53                | 30                          |
> | Cookie    | 200               | 60                          |
> | Spam      | 365               | 75                          |
> | Spray     | 548               | 95                          |
> | Lightbulb | 37                | 20                          |
> | Nutella   | 414               | 78                          |
> | Cube      | 120               | 35                          |
>
>
> We observe the expected positive correlation between mass and required current, which supports that the policy is indeed modulating force according to the identified mass.

---

> ### Author Response · Authors · 2025-11-22
>
> > **Q5**:Runtime analysis (Question 3)：
>
> **A5**: Regarding efficiency, we measured runtimes for each major module. For each object, our pipeline uses roughly 300–340 images. The end-to-end offline reconstruction (Structure-from-Motion via COLMAP plus 3DGS plus 2DGS) typically takes 30–35 minutes per object. Example statistics are:
> | Object  | # Images | COLMAP+3DGS+2DGS runtime | Mesh vertices |
> | ------- | -------- | ------------------------ | ------------- |
> | Lego    | 306      | 30 min                   | 47,266        |
> | Ketchup | 337      | 30 min                   | 13,487        |
> | Domino  | 320      | 30 min                   | 35,445        |
> | Cookie  | 309      | 30 min                   | 25,682        |
> | A       | 306      | 30 min                   | 67,895        |
> | U       | 327      | 33 min                   | 15,322        |
>
> Mass identification via differentiable physics scales with the number of mesh vertices. Each training iteration takes approximately 1.43–1.68 seconds, and convergence is typically reached within about 200 epochs. In practice, this corresponds to 5–20 minutes for most of the above objects.
>
>
> > **Q5**: Formatting issues (Suggestion 4)
>
>
> **A5**: Thank you for pointing these typos out. We have fix them accordingly.
>
>
> Once again, we thank the reviewer for these insightful comments. We hope that the additional experiments, quantitative evaluations, and clarifications described above can strengthen the completeness of our paper.
>
>
>
> [1] DRAWER: Digital Reconstruction and Articulation With Environment Realism, CVPR 2025
>
> [2] GSWorld: Closed-Loop Photo-Realistic Simulation Suite for Robotic Manipulation, arxiv 2025
>
> [3] GWM: Towards Scalable Gaussian World Models for Robotic Manipulation, RSS 2025
>
> [4] Robo-gs: A physics consistent spatial-temporal model for robotic arm with hybrid representation, ICRA 2025
>
> [5] SplatSim: Zero-Shot Sim2Real Transfer of RGB Manipulation Policies Using Gaussian Splatting, ICRA 2025
>
> [6] PhysGaussian: Physics-Integrated 3D Gaussians for Generative Dynamics, CVPR 2024
>
> [7] 4D Gaussian Splatting for Real-Time Dynamic Scene Rendering, CVPR 2024

---

> ### Author Response · Authors · 2025-11-27
> **Thank you for your reviews**
>
> Dear reviewer 6NV8,
>
> Thank you again for providing the very constructive review!
>
> Since the end of the rebuttal period is approaching (Dec 03, 9:00 PM UTC), we would like to kindly follow up to check if the provided responses have sufficiently addressed your questions and concerns. If so, we kindly hope that you might be willing to raise the level of your recommendation. Thanks again! :)
>
> Best, Authors of paper 333

---

### Official Review · Reviewer_snrT · 2025-11-03

**Soundness:** 3
**Presentation:** 3
**Contribution:** 3
**Rating:** 4
**Confidence:** 4

**Summary:**

This paper presents D-REX, a differentiable real-to-sim-to-real pipeline that couples Gaussian Splat Representations for photorealistic 3D reconstruction with a differentiable physics engine for object-mass identification and force-aware grasp policy learning. The system jointly optimizes physical parameters (mass) from robot interaction videos and learns manipulation policies conditioned on the inferred mass, closing the sim-to-real loop for dexterous grasping tasks.

**Strengths:**

1. The paper demonstrates a technically competent system that merges 3DGS and differentiable physics for vision-based grasping. Authors have performed real-world experiments validating some of their claims.

**Weaknesses:**

**1. Pipeline composition rather than a learning contribution.**
The full system is essentially a sequential pipeline: (1) Gaussian Splatting for 3D reconstruction with VLMs, (2) System identification to calibrate physical parameters, and (3) a procedural grasping policy that uses hand-designed grasp position and orientation heuristics. There is no novel algorithmic contribution or learning formulation that connects these modules beyond standard differentiable chaining.

**2. Hand-designed grasp prediction.**
The grasping procedure relies on manually defined rules. This is not significantly different from prior grasp pipelines that use geometry-based scoring or analytical quality metrics.

**3. No clear advantage over existing methods.**
The paper does not demonstrate how D-REX materially improves over existing differentiable grasping frameworks that already combine differentiable rendering and physics. The quantitative differences appear modest and could stem from tuning rather than a new principle. Additionally, just identifying mass, without taking materials into consideration, seems incomplete for robotics purposes.

**Questions:**

Please see my weaknesses section, thanks!

---

> ### Author Response · Authors · 2025-11-22
>
> We thank the reviewer for raising concerns about the degree of learning in our system, the role of hand-designed components, and the comparison to existing differentiable grasping frameworks. We address each point below.
>
> > **Q1**:Pipeline composition rather than a learning contribution.
>
> **A1**: Beyond a simple sequential pipeline: mass-aware system identification as the core novelty.
>  Our contribution lies beyond Gaussian Splatting, system identification, and a grasping policy, but the way we use differentiability to identify object mass and feed it back into dexterous grasp learning:
> First,  we introduce a trajectory-based loss that backpropagates through a differentiable simulator to identify an explicit mass parameter for each object. The identified mass is then injected into the dexterous policy as a physical latent, enabling the hand to modulate force and contact strategies for out-of-distribution weights.
>  As shown in Figure 5 and Figure 6 (referenced in the paper), grasps that succeed at nominal mass systematically fail when mass is significantly perturbed, unless the policy has access to an accurate mass estimate.
>
>
> Thus, the novelty lies in formulating and validating a mass-identification–conditioned dexterous grasping framework, rather than simply stacking existing modules.
>
> > **Q2**:Relation to hand-designed grasp prediction and geometry-based methods.
>
> **A2**:  We agree that many prior dexterous grasp pipelines rely on geometry-based scoring or analytical quality metrics. Our work is complementary but fundamentally different in two ways: Geometry-based policies such as UniDexGrasp++[1] use shape and contact geometry to score candidate grasps. As we show in our experiments, these methods struggle with very heavy objects: the same “good” geometric grasp can be unstable when the inertial properties deviate from what the policy implicitly expects.
> Analytical metrics in works such as Grasp’D[2] and DexGraspNet[3] focus on hand–object interaction in the geometric and contact domain. They do not explicitly estimate mass nor condition the policy on physical parameters, and thus cannot directly adapt to strong mass shifts.
>
>
> In contrast, our pipeline explicitly learns object mass from trajectory data and trains the dexterous hand to adapt its strategy across a range of inferred masses. Manually defined rules in our system are limited to simple pose/heuristic initializations; the key adaptation about how to close, squeeze, and stabilize under different weights is learned from data via the mass-aware policy.
>
> > **Q3**: Advantages over existing differentiable frameworks and focus on mass vs. other parameters.
>
> **A3**:  Regarding the claim that there is no clear advantage over existing differentiable grasping frameworks:  Empirically, we observe that baselines that do not perform explicit mass identification (including geometry-based scorers and differentiable-contact methods) degrade significantly when evaluated on out-of-distribution (OOD) masses. Our method maintains higher success rates and stability exactly in this regime, demonstrating a material gain attributable to mass-aware control, not just hyperparameter tuning.
>
>
> While there are differentiable frameworks that combine rendering and physics, like [4], to the best of our knowledge, there were no prior works that use Gaussian Splatting and the differentiable simulator for system identification like mass, from real executions and feed this estimate into a dexterous grasp policy to adapt to OOD scenarios.
> Finally,  we focus on mass because:
> In real-world settings, mass is a reliably measurable and accessible quantity in the real seetings, enabling quantitative evaluation of identification accuracy.
>
>
> Parameters such as friction and compliance are harder to measure and can vary with wear, temperature, or surface conditions. Our formulation, however, is general: the same trajectory-based identification framework could be extended to jointly optimize additional physical parameters in future work. In this paper, we deliberately isolate mass to demonstrate a clear, reproducible benefit from incorporating one well-defined physical parameter into dexterous grasp learning.
>
>
> In summary, our contribution is not merely pipeline composition, but a mass-aware, trajectory-driven system identification and learning framework that enables dexterous hands to adapt to significant variations in object mass—something that purely geometry-based or geometry-only differentiable methods do not explicitly address.
>
> [1] UniDexGrasp++: Improving Dexterous Grasping Policy Learning via Geometry-aware Curriculum and Iterative Generalist-Specialist Learning, ICCV 2023
>
> [2] Grasp'D: Differentiable Contact-rich Grasp Synthesis for Multi-fingered Hands, ECCV 2022
>
> [3] DexGraspNet: A Large-Scale Robotic Dexterous Grasp Dataset for General Objects Based on Simulation, ICRA 2023
>
> [4] One-Shot Real-to-Sim via End-to-End Differentiable Simulation and Rendering, RAL, May 2025

---

> ### Author Response · Authors · 2025-11-27
> **Thank you for your reviews**
>
> Dear reviewer snrT,
>
> Thank you again for providing the very constructive review!
>
> Since the end of the rebuttal period is approaching (Dec 03, 9:00 PM UTC), we would like to kindly follow up to check if the provided responses have sufficiently addressed your questions and concerns. If so, we kindly hope that you might be willing to raise the level of your recommendation. Thanks again! :)
>
> Best, Authors of paper 333

---

### Meta-Review · Area_Chair_on9N · 2026-01-06

**Summary:**

This paper proposes D-REX, a real-to-sim-to-real framework that combines 3D Gaussian Splatting with differentiable physics to identify object mass from video data and facilitate force-aware grasping. The reviewers generally praised the novelty of utilizing visual reconstruction for physics-based system identification and the method's effectiveness in bridging the sim-to-real gap. While one reviewer expressed concerns regarding the algorithmic novelty of the pipeline composition, the majority acknowledged the system's robustness and the significant performance gains over baselines. During the rebuttal, the authors effectively addressed outstanding technical concerns by providing quantitative evaluations of reconstruction quality, simulation details, and runtime analysis. Given the strong empirical validation and the practical value of the proposed system, the paper is recommended for acceptance.

**Reviewer Concerns:**

Addressed

1）Novelty (snrT): Clarified the unique contributions of the mass-aware feedback loop compared to geometry-only baselines.

2）Validation (6NV8): Added Real2Sim metrics (PSNR/SSIM), force-current correlations, and runtime analysis.

3）Theory (M426): Provided necessary ODE derivations and integrator ablations (Semi-implicit vs. Explicit).

Outstanding

1）Generalization (KAq7): Reliance on FoundationPose limits handling of small or symmetric objects.

2）Scope (snrT, KAq7): Identification is limited to mass, ignoring other dynamics like friction.

**Reviewer Scores:**

snrT: Positive (Predicted improvement)

6NV8: Positive

M426: Positive

KAq7: Positive

---

### Decision · Program_Chairs · 2026-01-26

Accept (Poster)